# A global estimate of monthly vegetation and soil fractions from spatio-temporally adaptive spectral mixture analysis during 2001-2022

Qiangqiang Sun[1], Ping Zhang[2], Xin Jiao[1], Xin Lin[1], Wenkai Duan[3], Su Ma[4], Qidi Pan[1], Lu Chen[1], Yongxiang Zhang[1], Shucheng You[5], Shunxi Liu[6], Jinmin Hao[1], Hong Li[7*], Danfeng Sun[1,8*]

[1]College of Land Science and Technology, China Agricultural University, Beijing, 100193,  China
[2]National Geomatics Center of China, Beijing, China.
[3]China Agricultural University Library, China Agricultural University, Beijing, 100193, China
[4]Chinese Research Academy of Environmental Sciences, Beijing, China
[5]Land Satellite Remote Sensing Application Center, Ministry of Natural Resources, Beijing, China
[6]China Land Survey and Planning Institute, Ministry of Natural Resources, Beijing, China
[7]Institute of plant nutrition and resources, Beijing Academy of Agriculture and Forestry Sciences, Beijing, China
[8]Technology innovation Center of land engineering, Ministry of Natural Resources, Beijing, China

*Correspondence to*: Danfeng Sun (sundf@cau.edu.cn); Hong Li (lih5176@126.com)

**Abstract.** Multifaceted regime shifts of Earth's surface are ongoing dramatically and—in turn—considerably alter global carbon budget, energy balance and biogeochemical cycles. Sustainably managing terrestrial ecosystems necessitates a deeper comprehension of the diverse and dynamic nature of multi-component information within these environments. However, comprehensive records of global-scale fractional vegetation and soil information that encompass these structural and functional complexities remain limited. Here, we provide a globally comprehensive record of monthly vegetation and soil fractions during the period 2001–2022 using a spatio-temporally adaptive spectral mixture analysis framework. This product is designed to continuously represent Earth's terrestrial surface as a percentage of five physically meaningful vegetation and soil endmembers, including photosynthetic vegetation (PV), non-photosynthetic vegetation (NPV), bare soil (BS), ice/snow (IS), and dark surface (DA), with high accuracy and low uncertainty, compared to previous vegetation index and vegetation continuous fields product, as well as traditional fully constrained linear spectral mixture models. We also adopt non-parametric seasonal Mann-Kendall tested fractional dynamics to identify shifts based on interactive changes of these fractions. Our results—superior to previous portrayal of the greening planet—not only report a $+9.35 \times 10^5$ km$^2$ change of photosynthetic vegetation, but also explore decrease of non-photosynthetic vegetation ($-2.19 \times 10^5$ km$^2$), bare soil ($-5.14 \times 10^5$ km$^2$), and dark surface ($-2.27 \times 10^5$ km$^2$). Besides, Interactive changes of these fractions yield multifaceted regime shifts with important implications, such as a simultaneous increase in PV and NPV in central and southwest China during afforestation activities, an increase of PV in cropland of China and India due to intensive agricultural development, a decrease of PV and increase of BS in tropical zones resulting from deforestation. These advantages highlight that our dataset which provides locally relevant information on multifaceted regime shifts at the required scale, enabling scalable modelling and effective governance of future terrestrial ecosystems. The data about fractional five surface vegetation and soil components are available on Science Data Bank (https://doi.org/10.57760/sciencedb.13287, Sun and Sun, 2023).

## 1 Introduction

Global terrestrial ecosystems are experiencing rapid and uncertain climate change and anthropogenic impacts since the twenty-first century (Alkama and Cescatti 2016; IPCC 2013; Song et al. 2018), which have profound impacts on shifts of Earth's surface, such as greening of the planet (Chen et al. 2019; Piao et al. 2006; Zhu et al. 2016), afforestation (Chen et al. 2019; Tong et al. 2018), deforestation (Qin et al. 2019; Zeng et al. 2018), agricultural expansion (Chen et al. 2019; Zeng et al. 2018; Yu et al., 2021), glacier melting (Hugonnet et al., 2021; Zemp et al., 2019; Soheb et al., 2022), and urban sprawl (Kuang et al. 2020; Liu et al. 2020; Zhang et al., 2022). These land surface shifts inversely play a fundamental role in affecting climate change via considerably altering the Earth's carbon budget, energy balance and biogeochemical cycles (Lawrence and Vandecar 2015; Qin et al. 2021). Increased understanding of these land cover changes is urgent requirement (Réjou-Méchain et al., 2021; Liu et al., 2020) to support the scientific, legislative and land management communities who strive to understand locally relevant knowledge and further protect, restore, and promote the sustainable use of terrestrial ecosystems under Sustainable Development Goal.

However, land surface interpretation is obstructed by extensive existence of mixed pixels in satellite imagery, especially in heterogeneous landscapes (Roberts et al. 1993). Continuous vegetation indexes (e.g., normalized difference vegetation index (NDVI), enhanced vegetation index (EVI)) provide limited information on surface composition, which hinders our ability of understanding ecosystem's structurally and functionally multifaceted shifts (Smith et al. 2019; Sun 2015; Zeng et al., 2023). In recent years, there have been significant advancements in fractional vegetation cover within the fields of remote sensing and environmental science. This progress has led to the development of various products at multiple resolutions, such as long-term global land surface satellite (GLASS), GEOV Fcover, multi-source data synergized quantitative remote sensing production system (MuSyQ) fractional vegetation cover (Baret et al. 2013; Jia et al., 2015; Mu et al., 2017; Zhao et al., 2023). These products primarily integrate and utilize data from different spectral bands and sensors, employing methods including machine learning, radiative transfer model and dimidiate pixel model (Baret et al. 2013; Yan et al., 2021; Zhao et al., 2023). However, these data primarily focus on green vegetation, posing significant limitations in capturing information regarding non-photosynthetic vegetation and bare soil. In ecological studies and remote sensing, non-photosynthetic vegetation including stems, branches, and other plant structures primarily serve functions other than photosynthesis, such as support and storage. Therefore, understanding the distribution and characteristics of non-photosynthetic vegetation is important for a comprehensive analysis of ecosystems and land cover, especially in drylands (Guerschman et al., 2009). Although some initiatives and products focused on multi-element fractions, such as MOD44B and the Global Vegetation Fractional Cover Product (DiMiceli et al., 2015; Guerschman et al., 2015). For instance, the Global Vegetation Fractional Cover Product primarily targets arid regions, particularly Australia, focusing on

photosynthetic vegetation, non-photosynthetic vegetation, and bare soil. Meanwhile, MOD44B achieves global-scale acquisition of trees, non-trees, and non-vegetative cover. There is a lack of unified classification systems among these products across global scale.

Previous advances in spectral mixture analysis method have facilitated investigation of estimating physically fractional vegetation and soil information in the mixed pixels with relatively few field points (Roberts et al. 1993; Small 2004; Smith et al. 1990). These unmixed endmember fractions provide multicomponent time series of information on surface heterogeneous composition and interactive evolution rather than individual vegetation indices (Elmore et al. 2000; Franke et al. 2009; Small and Milesi 2013; Sun 2015) and have been adopted to reveal the temporally dynamical systems under the influence of a changing environment and human activity (Lewińska et al. 2020; Suess et al. 2018; Sun et al. 2021). Recent studies have proven that spectral mixture analysis model has the advantage of providing more accurate and physically based representation of fraction vegetation-soil continues field in the subpixel level without training samples (Daldegan et al. 2019; Smith et al. 2007). This measurement offers a continuous, quantitative portrayal of land surface properties instead of discrete land cover classes, as well as superior to many of spectral indexes (e.g., vegetation index) (Rogan et al. 2002; Sun et al. 2019; Sun et al. 2020). Despite extensive validation and application of this method at the regional scale, there remain lack of global records of unmixed fractional vegetation and soil information, which may be resulted from the temporal and spatial variability of global intra-class and inter-class endmember spectra (Wang et al. 2021).

Recent advance in endmember variability has verified that Multiple Endmember Spectral Mixture Analysis (MESMA) was recommended be used in most applications considering its robustness in mitigating the endmember variability (Zhang et al., 2019). Such approach is well suited for heterogeneous landscapes because it allows an optimized model with varying the number and types of endmembers within each pixel (Roberts et al. 1998; Franke et al., 2009). However, considering world-wide landscapes with enormous heterogeneity under the climate fluctuations and human activities, the paradox of fine-grained spatial representation and challenged data processing for large scale and long-time series characterization of land surface has not yet been fully solved.

Here, we create a unified monthly fractional vegetation-soil nexuses product for the period 2001 to 2021, with an spatio-temporally adaptive MESMA methods at powerful Google Earth Engine (GEE) platform that provide powerful computational processing to realize planetary-scale analysis of geospatial data, at the same scale as monthly composites of MOD43A4 imageries (500×500m spatial resolution). This product is designed to continuously represent Earth's terrestrial surface as a percentage of surface endmembers with standard endmember spectra globally, providing a gradation of five surface vegetation and soil components: photosynthetic vegetation (PV), non-photosynthetic vegetation (NPV), bare soil

(BS), ice/snow (IS), and dark surface (DA). And we use non-parametric seasonal Mann-Kendall test to quantify global
trends and their interactive shifts in fractional vegetation-soil nexuses over the full period.
**2 Materials and methods**
**2.1 Dataset**
The MCD43A4 Version 6 Nadir Bidirectional Reflectance Distribution Function Adjusted Reflectance (NBAR) product
is selected in this study (Schaaf and Wang 2015). Since the view angle effects have been removed from the directional
reflectance, this dataset is provided as more stable and consistent daily surface reflectance imageries (bands 1-7) using best
representative pixel of 16-day retrieval period of Terra and Aqua spacecrafts at 500-m sinusoidal projection. The
MCD43A4 dataset was then temporally aggregated to produce a monthly composited dataset by taking the medium of all
valid reflectance in GEE platform during 2001–2022.
The Köppen-Geiger climate classification is a reasonable approach to aggregate complex climate gradients into a simple
but ecologically meaningful classification scheme (Beck et al. 2018). This dataset presents their widespread acceptance
and usage within the scientific community. This classification scheme includes five main classes and 30 subtypes (Beck et
al. 2018). We thus selected recently developed global Köppen-Geiger climate classification maps at a 1-km resolution for
the present-day (1980–2016). We initially used the 30-subtype classification for the selection of typical regions for the
endmembers collection. Meanwhile, we aggregated 30 sub-types to five main classes (i.e., tropical, arid, temperate, cold,
and polar) according to classification scheme criteria to represent a static climate condition in this study.
The land cover datasets are provided by the collection 6 MODIS land cover products (MCD12Q1) with 500-meter spatial
resolution in 2001 and 2022 (Friedl and Sulla-Menashe, 2015). MCD12Q1 utilizes multiple datasets and robust algorithms,
and provides detailed and reliable land cover information. It has been proven advantages in representing the global land
cover structure, patterns, and dynamics, aligning well with the requirements of our study for endmember selection. We
aggregate the International Geosphere-Biosphere Programme (IGBP) classification types of these datasets into three
regions—ecological zone, agricultural zone, urbanized zone. We define ecological zone as combination of evergreen
needleleaf forest, evergreen broadleaf forest, deciduous needleleaf forest, deciduous broadleaf forest and mixed forest,
closed shrublands, open shrublands, woody savannahs, savannahs, grasslands, permanent wetlands, Permanent snow and
ice, barren; refine agricultural zone as aggregation of cropland/natural vegetation mosaics; and represent urbanized zone
by urban and built-up lands.

## 2.2 Spatio-temporally adaptive spectral mixture analysis

Recent advances in spectral mixture analysis methods have facilitated investigation of estimating fractional endmember abundances in the mixed pixels (Meyer and Okin 2015; Okin 2007; Roberts et al. 1993). This method assumes that the reflectance of target mixing pixel is a linear combination of the weighting coefficients (proportional endmembers) and associated pure spectra,

$$R_i = \sum_{j=1}^{m} F_j E_{i,j} + \varepsilon_i \qquad (1)$$

Where $R_i$ is actual reflectance for band $i$; $E_{i,j}$ is the reflectance of a given endmember $j$ ($1 \leq j \leq m$) for a specific band $i$; $m$ is the number of endmembers; $F_j$ is fractional abundance of this endmember j; and $\varepsilon_i$ is the residual error for specific band $i$. The fully constrained least squares spectral mixture analysis model, including abundance sum-to-one constraint and abundance non-negativity constraint, is commonly applied for estimation of fractional endmembers to guarantee physically meaningful results (Heinz and Chein-I-Chang 2002). Spectral mixture analysis model is assessed by the model residual error ($\varepsilon_i$), reported as the root-mean-square-error ($RMSE_{sma}$), which can be expressed as Eq(2):

$$RMSE_{sma} = \sqrt{\frac{\sum_{i=1}^{n} \varepsilon_i^2}{n}} \qquad (2)$$

The spectral mixture analysis model includes three processes: endmember selection, and fraction estimation, and evaluation.

### 2.2.1 Nested endmember selection considering spatio-temporal variability.

The quality of spectral mixture analysis is significantly dependent on the representativeness of endmember selected. Endmember spectra used in spectral mixture analysis, in general, can either be derived from measured field spectral library or images (Franke et al. 2009; Sonnentag et al. 2007). The image-based endmember selection method is more practical way because advantage of image endmember is that they can be collected at the same scale as the image and are relatively easy to associate with image features (Rashed et al. 2003). Given that such endmember selection would be hampered by temporal and spatial variability of global intra-class and inter-class endmember spectra, we develop a nested framework for endmember selection considering spatial and temporal variability (Fig. 1).

(1) Recent studies have proposed various compositional endmember frameworks in different application contexts. For example, a framework including substrate, vegetation, dark and ice/snow was proposed and verified globally for both Landsat and MODIS to allow estimated fractions, this framework ensures consistent comparison of estimated fractions across diverse climate patterns and land cover types (Small and Milesi 2013; Sousa and Small 2019). Another framework

includes photosynthetic vegetation, non-photosynthetic vegetation, soil, and shade (Roberts et al. 1993), this framework was widely adopted for presentation surface structure worldwide, particularly in tropical rainforest and dryland ecosystems (Guerschman et al., 2015). These elements can characterize the fundamental composition of the Earth surface. Thus, we embody five endmembers to represent surface units, these five endmembers include photosynthetic vegetation (PV), non-photosynthetic vegetation (NPV), bare soil (BS), dark (DA), ice/snow (IS). Concretely, PV refers to green photosynthetic foliage characterized by chlorophyll absorptions in the visible and high reflectance in the near-infrared bands; NPV represents non-tilled cropland/grassland, and tree litters; BS contains soil, rock, and sediment. DA represents a fundamental ambiguity; thus, it may be either absorptive (e.g., black lava), transmissive (e.g., deep clear water) or non-illuminated (shadow) surface. IS is permanent glaciers and snow that are widespread in the polar regions and high mountains.

(2) Considering both climate patterns and land cover types, the typical sites employed for standardized endmembers selection were chosen based on global MODIS sinusoidal grid (10°× 10°intervals). The Köppen-Geiger climate classification zones is adopted as the dominant criterion to undertaking full coverage of climate types (Beck et al. 2018). Meanwhile, we also examine land cover diversity, characterized by Simpson's Diversity Index (D) of recent MCD12Q1 Version 6 product in 2020 in each MODIS grid.

$$D = 1 - \sum_{i=1}^{m} (P_i)^2 \tag{3}$$

Where $P_i$ is percentage of type $i$ land use and cover in the grid, $m$ is number of land use and cover in the grid. Finally, we selected the top 10 grids (i.e., h08v05, h12v12, h13v09, h16v01, h21v03, h22v02, h22v08, h24v06, h26v05, h27v06, h29v12) in terms of Simpson's Diversity Index (D) among all MODIS grids (Fig. S1a, b), and containing all Köppen-Geiger climate types (Fig. S1c), were selected for generation of standardized endmember spectrum.

(3) The representativeness of endmembers always shifts with time variation. A multi-temporal endmembers selection scheme has been validated for various time series images (Sun and Liu 2015; Sun et al. 2018). This process of utilization of both spatially and temporally mixed image collections for endmember selection can consider both spatial and temporal variability. Therefore, the multi-temporal standardized endmembers selection scheme is adopted in 10 typical zones that considering both climate and land cover diversity. Principal component (PC) transformation derived eigenvectors and associated PC images were utilized as criteria for determination of endmember types. Specifically, eigenvector of PC, displaying remarkable differences between shortwave infrared bands with other visible and near-infrared bands, is obviously able to highlight characteristics of IS. PC eigenvector with relatively high contrast between the near-infrared band and other bands primarily captures information related to photosynthetic vegetation (PV), particularly during vegetation growing seasons. The BS and NPV will be boosted with the PC when corresponding eigenvector emerges the

same direction. Even though there is no obvious regular pattern of eigenvector for DA determination, the PC images can provide adequate information coupled with high-resolution images of Google Earth. After the determination of endmembers type and their PCs in each grid, we ranked these PCs by descending order of the variance contribution, and selected PC images of first three timings for endmember selection. We have listed the endmember types and their highlighting timings for each selected gird in Table S1. The image endmembers can be acquired from the vertex's pixels (200-400 pixels) of scatter plot formed by the PC images at their corresponding timings in each grid. We then exported these acquired pure pixels as regions of interest to compute original MODIS reflectance as endmember spectra. These selected pure pixels for each endmember are validated by high spatial resolution remote sensing imagery of Google Earth (Fig. S2).

(4) Besides, we collect MODIS derived endmember spectra used in previous study to complement and enrich the diversity of the spectral library (Okin et al. 2013; Daldegan et al. 2019; Meyer and Okin 2015; Sousa and Small 2019). We gather 7 PV, 5 NPV, 5 BS, and 1 DA endmember spectra through such literature search method. Finally, we establish a library of endmember spectra considering spatio-temporal variability, this library includes 35 PV spectra, 40 BS spectra, 25 NPV spectra, 16 DA spectra, and 15 IS spectra.

(5) To ensure feasibility of pixel-by-pixel operations in GEE, we also consider the similarity between the spectral curves, the hierarchical clustering method is selected to aggregate these spectra of each endmember as sub-groups, we input all spectral curves per endmember, grouping similar curves to compute their mean—a representative typical spectral curve for each cluster. Such hierarchical clustering boasts strong interpretability and adaptability for clustering at diverse scales within data analysis. Finally, we obtain 4 PV spectra, 4 BS spectra, 3 NPV spectra, 2 DA spectra, and 2 IS spectra to estimate vegetation and soil fractions at global scale during 2001 to 2020 (Fig. 2).

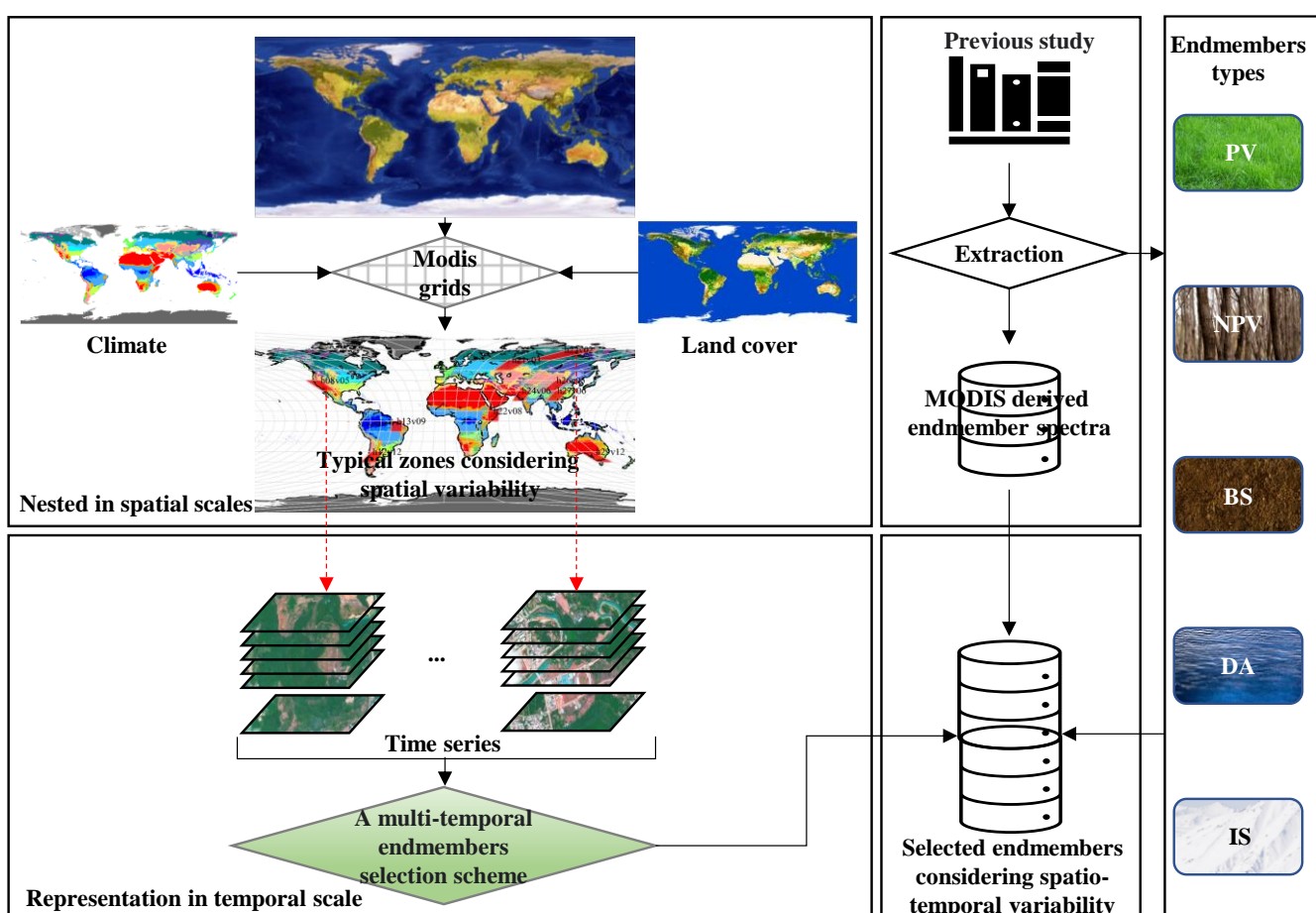

220

**Figure 1: A framework for endmember selection considering spatial and temporal variability.**

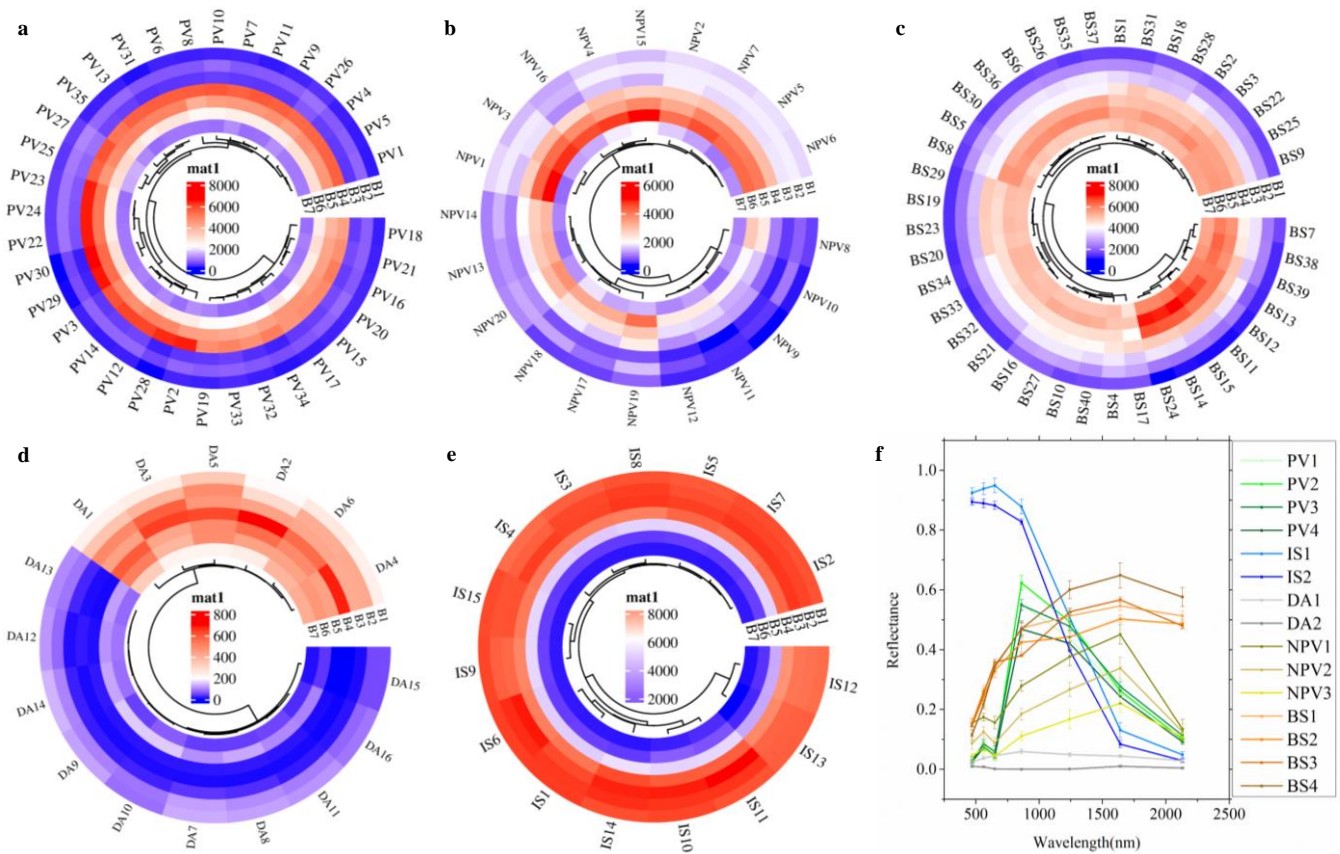

**Figure 2: Endmember spectra. a-e**, Hierarchical clusters of the endmember spectra of PV, NPV, BS, DA and IS. **f**, the averaged final endmember spectra including 4 PV spectra, 4 BS spectra, 3 NPV spectra, 2 DA spectra, and 2 IS spectra. B1-B7 represent MODIS spectral bands, including 459-479nm, 545-565nm, 620-670nm, 841-876nm, 1230-1250nm, 1628-1652nm, and 2105-2155nm.

### 2.2.2 Multiple Endmember Spectral Mixture Analysis

The MESMA has been used to estimate fractional vegetation-soil nexuses based on selected endmember spectra. According to the convex geometry concepts, the number of endmember (n+1) in the model should be equal to the intrinsic dimensionality of the spectral space (n) plus one (Boardman 2013). We found the cumulative contribution of the top three PCs has exceeded 99% (Fig. S3), this three-dimensional PC space allows four-endmember models. We initially generate multiple endmember combinations based on selected endmember spectra, and achieve 692 combination models, including two-endmember model (88), three-endmember model (252) and four-endmember model (352) (Table S2). The fully

constrained least squares spectral mixture analysis model is selected to estimate fractions and count $RMSE_{sma}$ for each
endmember combination in GEE platform. We finally search a specific endmember combination with the smallest
$RMSE_{sma}$ and achieve the estimated endmember fractions of this combination as final fractions.

### 2.3 Direct validation of the dataset

The smallest $RMSE_{sma}$ of 692 combination models is adopted as criteria to assess suitability and uncertainty of the model.
The model suggests a generally good fit when mean $RMSE_{sma}$ over the image is less than 0.02 (Wu and Murray 2003).
Moreover, due to challenges in conducting fraction estimation validation through field surveys, we employ reference data
obtained from high spatial resolution images as validation set. We thus select for two sets of reference data that their land
cover classification systems are closely related to our five endmembers. Global Land Cover Validation Reference Dataset
(GLCVRD) is provided with a 2m reference dataset from very high resolution commercial remote sensing data within 5 ×
5 km blocks from 2003 to 2012 (Olofsson et al. 2012; Pengra et al. 2015; Stehman et al. 2012). These datasets support
global estimates of classification accuracy for four major land cover classes: tree, water, barren, other vegetation, cloud,
shadow, ice & snow. Various recent studies have selected this dataset to evaluate the continuous fields of land cover types
(Baumann et al. 2018; Qin et al. 2019; Song et al. 2018). We use all GLCVRD reference dataset (Fig. 3a) to assess the
accuracy of globally fractional vegetation and soil estimates from MESMA. Firstly, we filter the estimated fractions based
on the corresponding year and month obtained from the reference data. Simultaneously, aligning the interpretations of land
cover types with our endmembers, we pair them accordingly, that is, tree and other vegetation represent PV and NPV,
barren stands for BS, water and shadow correspond to DA, and ice & snow denote IS. Subsequently, we reclassify these
paired land cover types and calculated their percentage within 5×5 km blocks, in which we exclude cloud coverage (named
no data). Additionally, utilizing these cloud-free pixels in each block, we compute the mean of fractional values for each
endmember, and then compare these estimated fractions with the measured percentage of paired the reclassified land cover
types to validate the reliability of our product (Fig. S4). Based on paired measured fractions and our estimated fractions
within blocks, we adopt four accuracy metrics including mean error (ME), mean absolute error (MAE), root-mean-square-
error (RMSE), and $R^2$ for accuracy assessment. ME measures the average of all errors in the dataset where errors are the
differences between predicted and actual values, MAE calculates the average of the absolute differences between predicted
and actual values, RMSE provides a measure of prediction error, whereas $R^2$ offers insight into the amount of variability
in the dependent variable that the model explains. These metrics provide a more comprehensive assessment of the model's
accuracy, helping to understand different facets of its performance, such as bias, variability, and overall predictive power
(James et al., 2013).

$$ME = \frac{\sum_{i=1}^{n}(p_i - r_i)}{n} \tag{4}$$

$$MAE = \frac{\sum_{i=1}^{n}|p_i - r_i|}{n} \tag{5}$$

$$RMSE = \sqrt{\frac{\sum_{i=1}^{n}(p_i - r_i)^2}{n}} \tag{6}$$

$$R^2 = 1 - \frac{\sum_{i=1}^{n}(p_i - r_i)^2}{\sum_{i=1}^{n}(p_i - \bar{r})^2} \tag{7}$$

Where $p_i$, $r_i$ are estimated endmember fractions and reference endmember fractions at $i$th block, $n$ is sample size ($n = 474$), $\bar{r}$ is mean of the reference endmember fractions of all blocks.

Besides, we also authenticate our product through incorporating comprehensive global land cover and land use reference data (Fritz et al. 2017), which were obtained from the Geo-Wiki crowdsourcing platform across four campaigns: Human impact, wilderness, reference and disagreement. Over 150000 samples of land cover and land use were acquired in this reference data. To effectively validate our product, we need to filter the reference data, considering aspects such as data acquisition time, measurement methods, and credibility. We select first three campaigns, which have a good match with MODIS pixels (size 1×1km) and were observed during 2001 to 2022. High feasibility reference data is then selected through the confidence information of land cover estimates and the status of use of high spatial resolution imagery provided by the metadata. Similarly to the procedural description used for fractional vegetation-soil compared to GLCVRD, we reclassify ten classes of this dataset into our four groups of endmembers, including (1) tree cover, shrub cover, herbaceous vegetation/grassland, cultivated and managed, and mosaic of cultivated and managed/natural vegetation to PV and NPV; (2) flooded/wetland and open water to DA; (3) urban and barren to BS; (4) snow and ice to IS. This involve comparing the measured percent of land cover with the mean of endmember fractions within the corresponding 1×1km pixels.

**2.4 Comparisons and uncertainties analysis**

To verify the consistency and merits of our dataset against existing ones, we conducted comparisons with four distinct pre-existing datasets: NDVI, Leaf area index (LAI), MOD44B Vegetation Continuous Fields product, GLASS fractional vegetation cover dataset, and GEOV Fcover dataset. NDVI is derived from monthly synthesized MCD43A4 images. LAI is derived from 8-day composite MOD15A2H V6 dataset at 500m resolution. Both mean values of NDVI/LAI and our estimated fractional PV across all years are considered for comparison. The MOD44B Vegetation Continuous Fields

product provides annual information about the percent tree cover, percent non-tree cover, and percent non-vegetated within
each 250-meter pixel globally (DiMiceli et al., 2015). Consequently, we compare vegetation cover proportions—sum of
percent tree cover and percent non-tree cover—to the sum of fractional PV and NPV. To align spatial and temporal
resolutions, we aggregated the sum of percent tree cover and percent non-tree cover to a 500-meter scale. Simultaneously,
we computed monthly fractional PV and NPV as annual averages. The GLASS fractional vegetation cover dataset, offering
an 8-day temporal frequency and dual spatial resolutions of 0.05° and 500 meters, was generated using a machine learning
approach correlating MODIS reflectance with fractional vegetation cover (Jia et al., 2015). In our study, the 500-meter
GLASS data was utilized to validate our estimated fractions. We computed annual averages from all the CLASS fractional
vegetation cover data within a year and compared it with the annual averages of Fractional PV and NPV. GEOV FCover
is a 10-day product estimated through the neural network using visible, near-infrared and shortwave infrared at 1km
resolution (Baret et al. 2013). We aggregate our product to a 1km spatial resolution, and compare their annual averages
with the annual averages of GEOV FCover.
Moreover, we also carry out a comparison with traditional linear spectral mixture analysis to demonstrate the advantages
of our spatio-temporally adaptive spectral mixture analysis. Such comparison is performed using average of monthly
$RMSE_{sma}$ of fully-constrained framework based on two fixed endmember spectral curves: (1) average of all spectral
spectra for each endmember and (2) existing spectral spectra from Small and Sousa (2019).
Furthermore, to validate the uncertainties of the hierarchical clustering, we select a spectral spectrum from selected
endmember spectra that exhibit the largest mean squared error from the mean of cluster for each cluster. These selected
spectral spectra were then used to reconstruct an extreme library of endmember spectra and used to estimate fractional
vegetation and soil using MESMA.
**2.5 Change of vegetation and soil fractions**
Mann-Kendall test is commonly referred to as a nonparametric test method, which is procedures that detects monotonic
trends of sequences over time (Kendall 1975; Mann 1945; Bradley 1968). When seasonal environmental data of interest
are available as time series for which the time intervals between adjacent observations arc less than one year (i.e., daily,
weekly, and monthly sequences), a multivariate extension of the Mann-Kendall test has been advanced to handle seasonal
sequences. Besides, the seasonal Sen's slopes (change per unit of time) are commonly chosen to express this magnitude
(Hirsch et al. 1982; Sen and Kumar 1968). Therefore, we impose the seasonal Mann-Kendall test and seasonal Sen's method
to define trend and slope (annual change) of endmember fractions at the pixel level. The detailed information about
seasonal Mann-Kendall test and seasonal Sen's method can be found in Supplementary Methods. If the Mann-Kendall test
is not statistically significant ($p \geq 0.05$), we define net change as 0. If the trend test is significant ($p < 0.05$), we apply the
seasonal Sen's method to estimate the per-pixel net change between 2001 to 2022 (i.e., slope times 22 years). Besides, we
aggregate per-pixel net change of endmember fractions to spatial scales (such as country, biome, climate zone) to obtain
total area change estimates at these aggregated scales from 2001-2022 as,
$$\text{Net area change} = \sum_{i=1}^{n} T_i A_i N \tag{8}$$

Where $T_i$ is Sen's slope of endmember fraction for a statistically significant pixel $i$, $A_i$ represent area of pixel $i$, $n$ is the
total number of such pixels in the region, $N$ is the length of study period ($N = 22$).

## 3 Results

### 3.1 Evaluation of monthly estimates of vegetation and soil fractions

We utilize standard endmember spectra globally to estimate fractional vegetation-soil nexuses via MESMA. The simulated
results elucidate that the MESMA model performs well with an ideal model $RMSE_{sma}$ over globe ($0.018\pm0.022$, Fig. 3a-
c). We find the regions with $RMSE_{sma}$ above 0.02 account for less than one-fifth of the global area and are mainly
distributed in barren such as Sahara Desert and polar regions. This exceptional performance demonstrates the superiority
and low uncertainty of the model. This performance is also evidenced by evaluation results from GLCVRD (Fig. 3e-h,
Table S3). Specifically, the performance of PV+NPV, BS, and IS endmember estimates have MAE less than 0.118, RMSE
less than 0.149, $R^2$ greater than 0.592. Although the MAE (0.050) and RMSE (0.065) perform well, the $R^2$ of estimated
DA against measured DA presents only 0.156, largely attributed to the absence of estimations for shadows cast by smaller
vegetation within the validation dataset. In blocks with a DA greater than 0.2, the estimated DA and measured DA present
better consistency, in which the shadows of hills are well measured by GLCVRD. Moreover, we simultaneously select
another set of land cover reference data as validation samples (Fig.S5). The validation results demonstrate the superiority
of our estimation products, with MAE for PV, NPV, DA, BS, and IS abundances all less than 0.099, RMSE (Root Mean
Square Error) all less than 0.129, and $R^2$ all greater than 0.57. However, this set of validation data is also not ideal as it
fails to accurately estimate small-scale vegetation shadows and bare soil in highly vegetated area, resulting in a slight
overestimation of our DA and BS estimates near zero, accompanied by an underestimation of PV and NPV in high-value
areas.

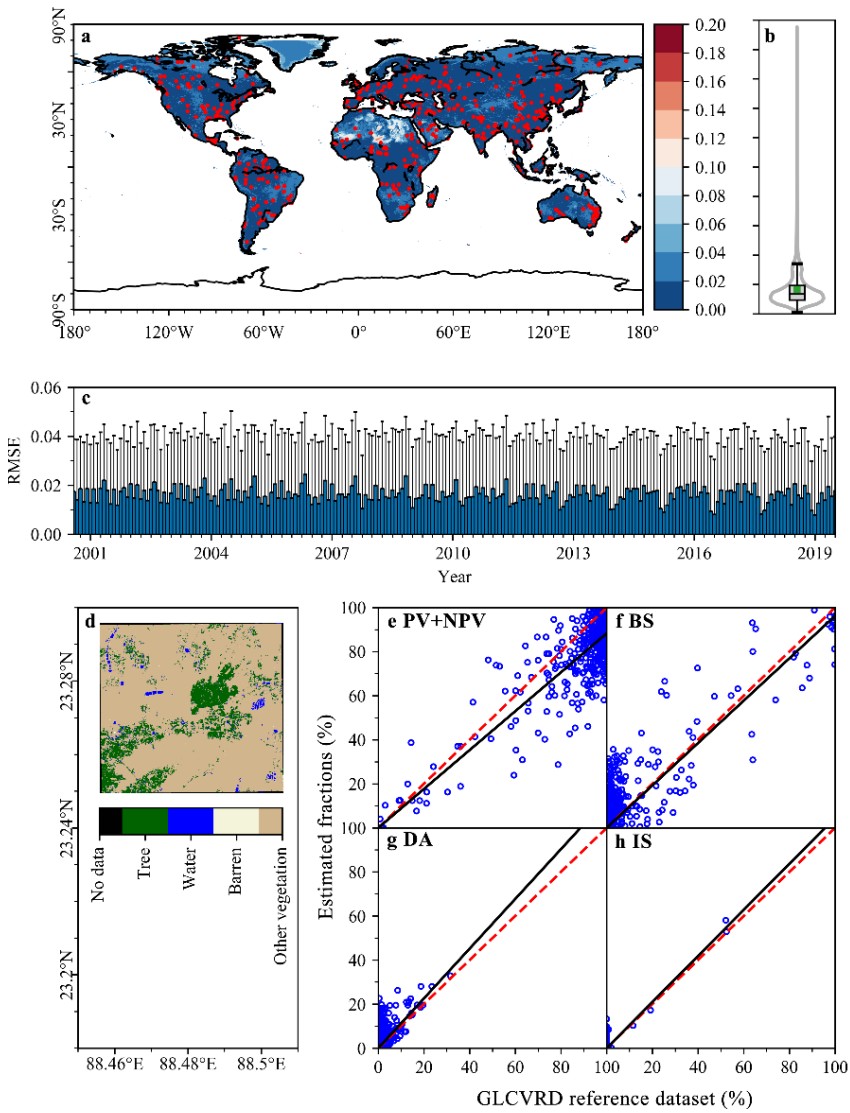


**Figure 3: Evaluation of global fractional endmember estimates**. **a**, the spatial pattern of average of monthly $RMSE_{sma}$ from 2001 to 2022, the overlaid red dots were spatial distribution of the 5 × 5 km validation blocks of GLCVRD reference dataset (n=500). **b**, the boxplot and violin plot for average of monthly $RMSE_{sma}$ (a), which indicate mean $RMSE_{sma}$ over image is less than 0.02. **c**, monthly averaged $RMSE_{sma}$ from 2001 to 2022 with error bars. **d**, the schematic of detailed land cover classes of GLCVRD reference dataset. **e-h**, Scatter plots of PV+NPV, BS, DA, IS fractions against GLCVRD

reference dataset (tree + other vegetation, barren, water + shadow, ice & snow). Endmember fractions were derived from
corresponding year and month of each $5 \times 5$ km block achieved.

**3.2 Compared with other datasets and traditional spectral mixture analysis model.**

We compare our estimates vegetation and soil fractions dataset with NDVI, fractional PV and NPV against fractional tree
and non-tree vegetation of MOD44B vegetation continuous fields product and other fractional vegetation cover products.
We detected a strong positive relationship between PV fraction and NDVI. Yet, this correlation becomes less pronounced
when PV exceeds 50%, suggesting an evident saturation effect within NDVI (Fig. 4a). This linear relationship also exists
in the relationship between PV and LAI, but a non-linear turning point occurs when PV exceeds 70% (Fig. 4c). Furthermore,
PV and NPV fraction displays a significant positive association with the remaining three fractional vegetation cover
products (Fig. 4b, d, e). Specifically, the MOD44B vegetation continuous fields product reveals an $R^2$ of 0.75 with a p-
value below 0.01, the GLASS product displays an $R^2$ of 0.69 with a p-value below 0.01, and the GEOV Fcover product
exhibits an $R^2$ of 0.65, also with a p-value below 0.01. Nevertheless, within regions with lower vegetation cover, especially
drylands that present a higher presence of non-photosynthetic materials, current products (particularly GLASS and GEOV
Fcover) have not adequately evaluated vegetation coverage, resulting in some degree of underestimation in the outcomes
(Fig. 4c, d, Fig. S6a). Furthermore, we notice overestimation in the MOD44B vegetation continuous fields product in areas
where vegetation cover is less than 50%, mainly due to insufficient estimation of dark components (i.e., shadow of
vegetation and mountain, water) (Fig. 4b, Fig. S6c). In areas with denser vegetation cover, we found good alignment among
these products, especially with the MOD44B vegetation continuous fields product. However, the GLASS and GEOV
Fcover products tend to underestimate certain areas, primarily focusing more on green vegetation and overlooking non-
photosynthetic components (Fig. 4c, d, Fig. S6b). Moreover, both of two fully constrained linear spectral mixture models
are inferior to our framework since we consider the variability of the spectra in both time and space (Fig. 4e, f).

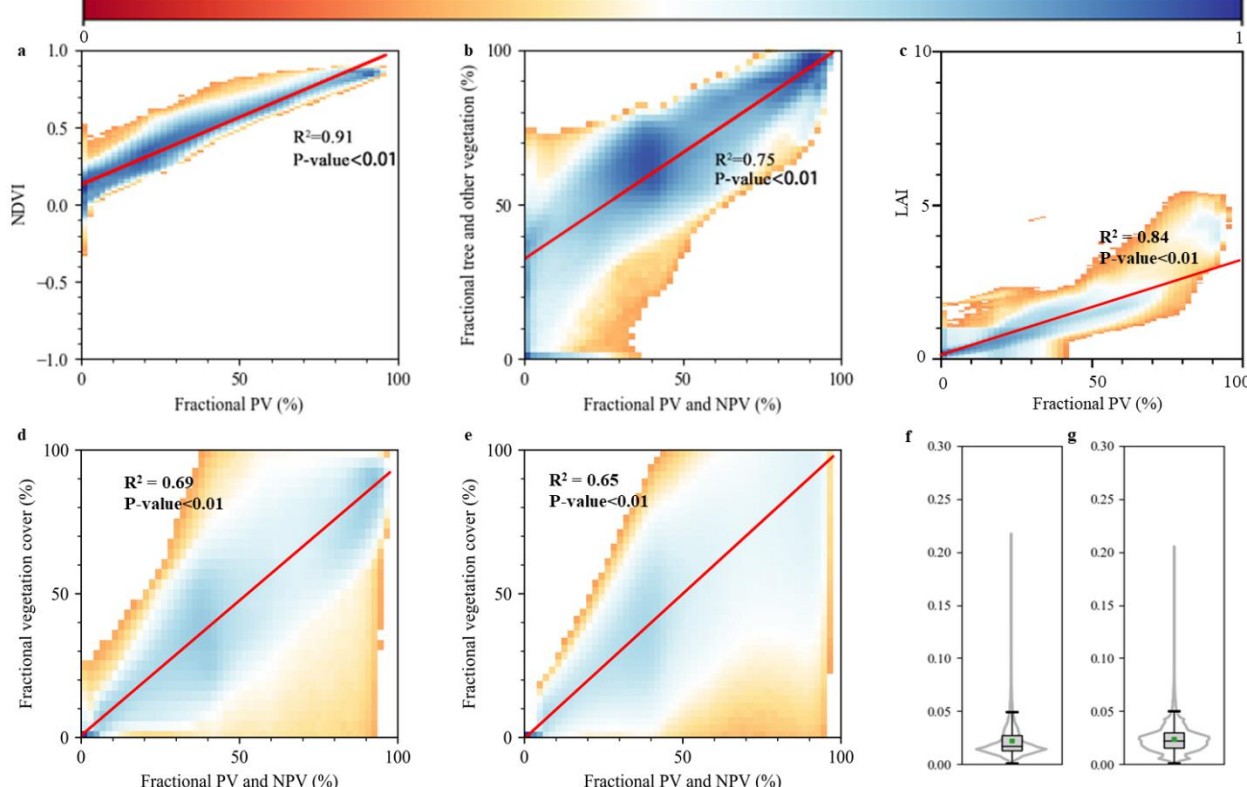


**Figure 4: Comparisons with other datasets and traditional spectral mixture analysis models. a, b**, **c**, **d, e,** the bi-dimensional histogram of fractional endmembers and other dataset with bin size of 2%, including fractional PV against NDVI (a), fractional PV and NPV against fractional tree and non-tree vegetation of MOD44B vegetation continuous fields product (b), fractional PV against LAI (c), fractional PV and NPV against GLASS fractional vegetation cover product (d), fractional PV and NPV against fractional vegetation cover of GEOV Fcover product (e); **f, g,** the boxplot and violin plot for average of monthly $RMSE_{sma}$ for two fixed endmember spectral curves using fully constrained linear spectral mixture models, including (e) average of all spectral spectra for each endmember and (f) existing spectral spectra from Small and Sousa (2019).

## 3.3 Uncertainties of estimates of global vegetation and soil fractions

It can be found that 90% of the $RMSE_{sma}$'s differences are concentrated within 1% (Fig. 5a), indicating the relative stability of the unmixed results from two libraries as well as the effectiveness of the clustering. These are also corroborated by the

differences between unmixed endmember fractions (Fig. 5b-e), as indicated by that more than 90% of global pixels have a
difference of 10% or less, as well as more than 70% of global pixels present a difference up to 1%, except for the two
endmembers with higher spatial variability (NPV, 61.59%; DA, 62.59%).

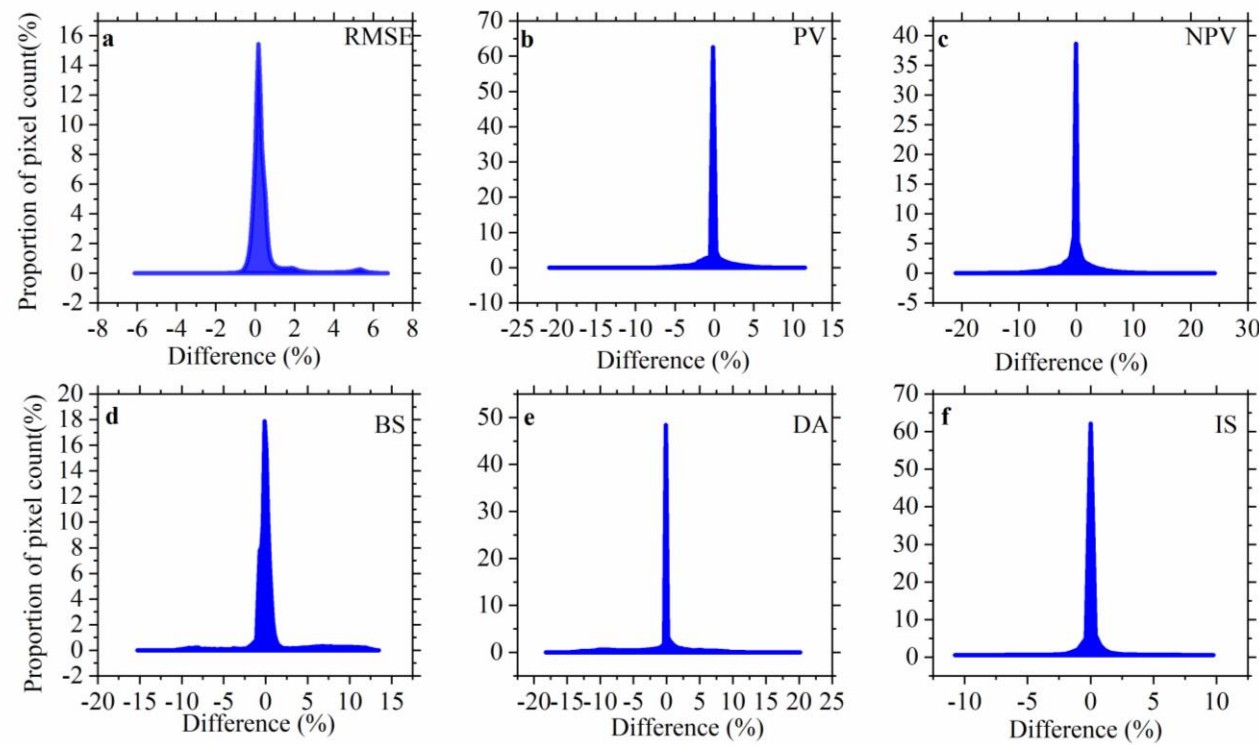


**Figure 5: Difference in unmixed results between mean endmember library and endmember library in hierarchical**
**cluster. a, b, c, d, e and f** represent histogram of $RMSE_{sma}$, PV, NPV, BS, DA and IS.
**3.4 Spatial distribution of global vegetation and soil fractions**
Globally averaged monthly gradations of five surface vegetation and soil components are illustrated in Fig. 6. Our estimates
depict that PV cover presents the largest area for both 30°-60°N and 0-30°S, which together account for more than half of
the total global terrestrial vegetation area. We find the average PV fraction in the Northern Hemisphere is significantly less
than that in the Southern Hemisphere, especially in the Amazon, although the area of PV at 30°-60°N is slightly greater
than that of 0-30°S. Dominated by foliage-free desert vegetation and agricultural straw, NPV is mainly found in the semi-
arid regions (e, g., USA, western China, and Australia) and croplands. BS is also located in the drylands of the Sahara,
western Asia, and west-central Australia in terms of both fraction and total area. DA and IS, on the one hand, are mainly

concentrated in in terrestrial water bodies and mountains, Greenland and global high mountains of the Himalayas and the Andes, respectively.

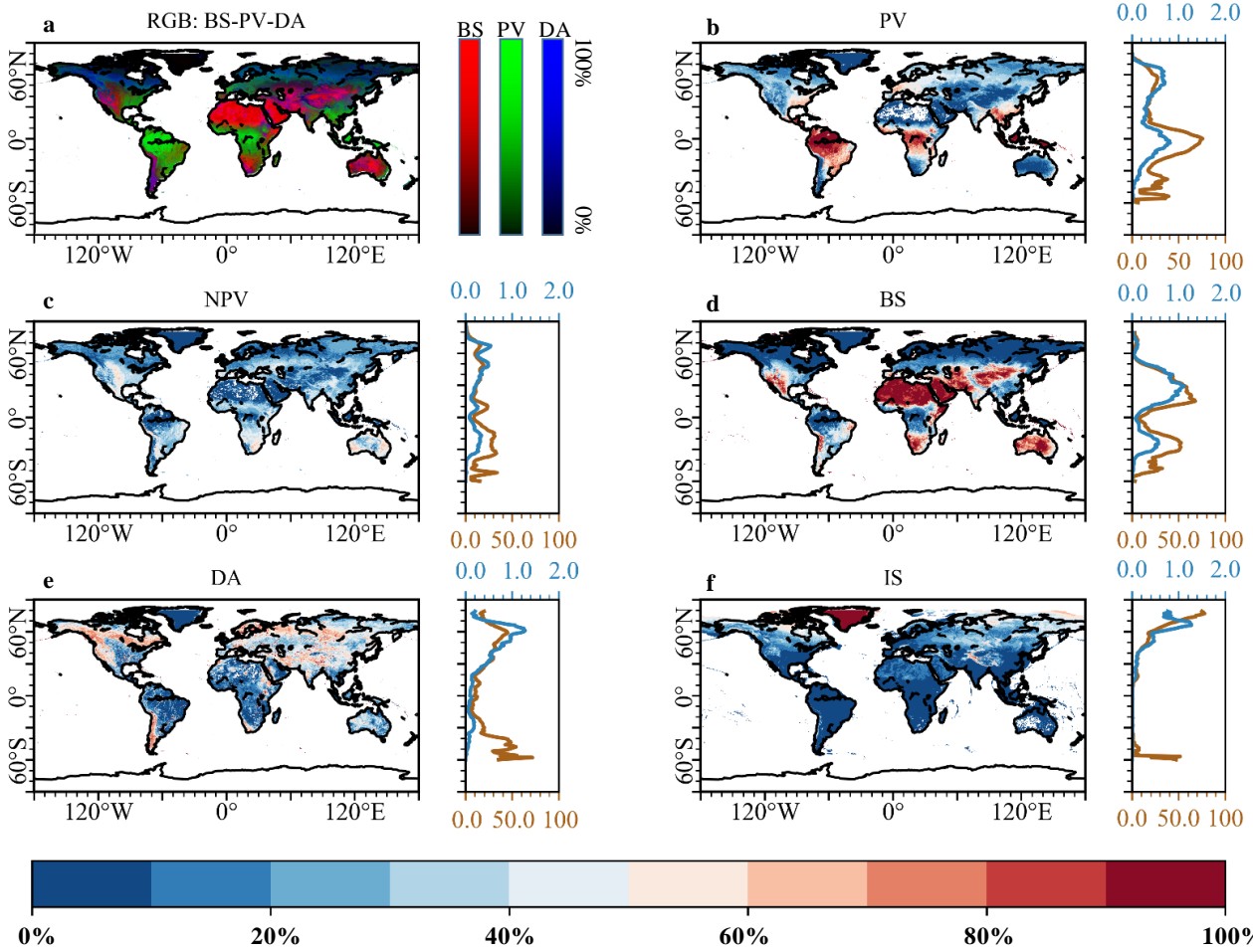

**Figure 6: Global average of monthly fractional endmembers from 2001 to 2022**. a, Spatialized RGB composition of three averages of monthly fractional endmembers (RGB: BS-PV-DA). b-f, average of PV, NPV, BS, DA, and IS fractions. Shadowed subplots are average of fractional endmembers (%, orange, lower) and area of endmembers (fraction × pixel area, ×10$^6$ km$^2$, blue, upper) at respective latitudes, taking each degree as the statistical standard.

**3.5 Globally and regionally fractional endmembers dynamics**

The total area of PV increases $9.35\times10^5$ km$^2$ from 2001 to 2022, which represents a +1.88% change relative to 2001 green vegetation (Fig. 7; Table S4). This increased trend results from higher magnitude of gain ($1.57\times10^6$ km$^2$), nearly 2.5 times the loss area. Our PV area gain estimate basically agrees in magnitude with the global vegetation continuous fields product's estimate of net vegetation area change ($1.36\times10^6$ km$^2$), despite differences in the time period covered (1982-2016) and definition (tree and other vegetation) (Song et al. 2018). Temperate, arid and cold regions together contribute more than 90% of the greening area (Fig. 8; Table S4). In these areas, the China and India are two major contributors (Fig. S7) through land use management like ecological afforestation and agricultural expansion (Chen et al. 2019). Within Brazilian Amazon, we find a large area of PV loss (Fig. S7), which is also supported estimates of forest cover and loss (Qin et al. 2019).

A decreasing trend is observed in NPV globally ($2.19\times10^5$ km$^2$), representing a -1.45% change relative to 2001 NPV area (Fig. 7; Table S4). Tropical and temperate regions together contribute more than 80% of the loss area of NPV, which may result from global warming induced tree greening. Although the arid is major source of NPV ($2.75\times10^6$ km$^2$ in 2001, 18.2% of globe NPV area), the change area of NPV is only less than 10000 km$^2$ (Fig. 8; Table S4).

In the context of the greening of the vegetation, the degree of BS is reduced by $5.14\times10^5$ km$^2$ during study period, indicating a -1.09% change relative to initial BS of 2001. The decreased global BS trend occurs in temperate, arid and cold regions, accounting over 90% of net BS change area. In contrast, tropical region appears an increasing trend (+$1.22\times10^5$ km$^2$), and thus offset the decline in BS in the rest of the regions (Fig. 8; Table S4). This outcome results from the forest loss induced soil exposure in Brazilian Amazon and Southeast Asia (Fig. S7). Meanwhile, the total area of DA also represents a net change of -$2.27\times10^5$ km$^2$, from 2001 to 2022, which represents a -0.69% change relative to 2001 DA area. The largest negative contributions to the decreased global DA appear in cold (46.26%) and arid (32.87%) (Fig. 6; Table S4). We observed an increase of $2.46\times10^4$ km$^2$ in IS globally, which represents a +0.11% change relative to 2001 IS. Such positive trend is mainly benefited by the increase of snow and ice in the cold regions, in which the net increase area is 1.5 times

greater than the global net IS change (Fig. 8; Table S4). This is caused by the increase of snowfall. However, global
warming is causing a substantial melting of snow and ice, resulting in the arid, tropical, temperate and polar regions show
a decreasing trend in IS cover.

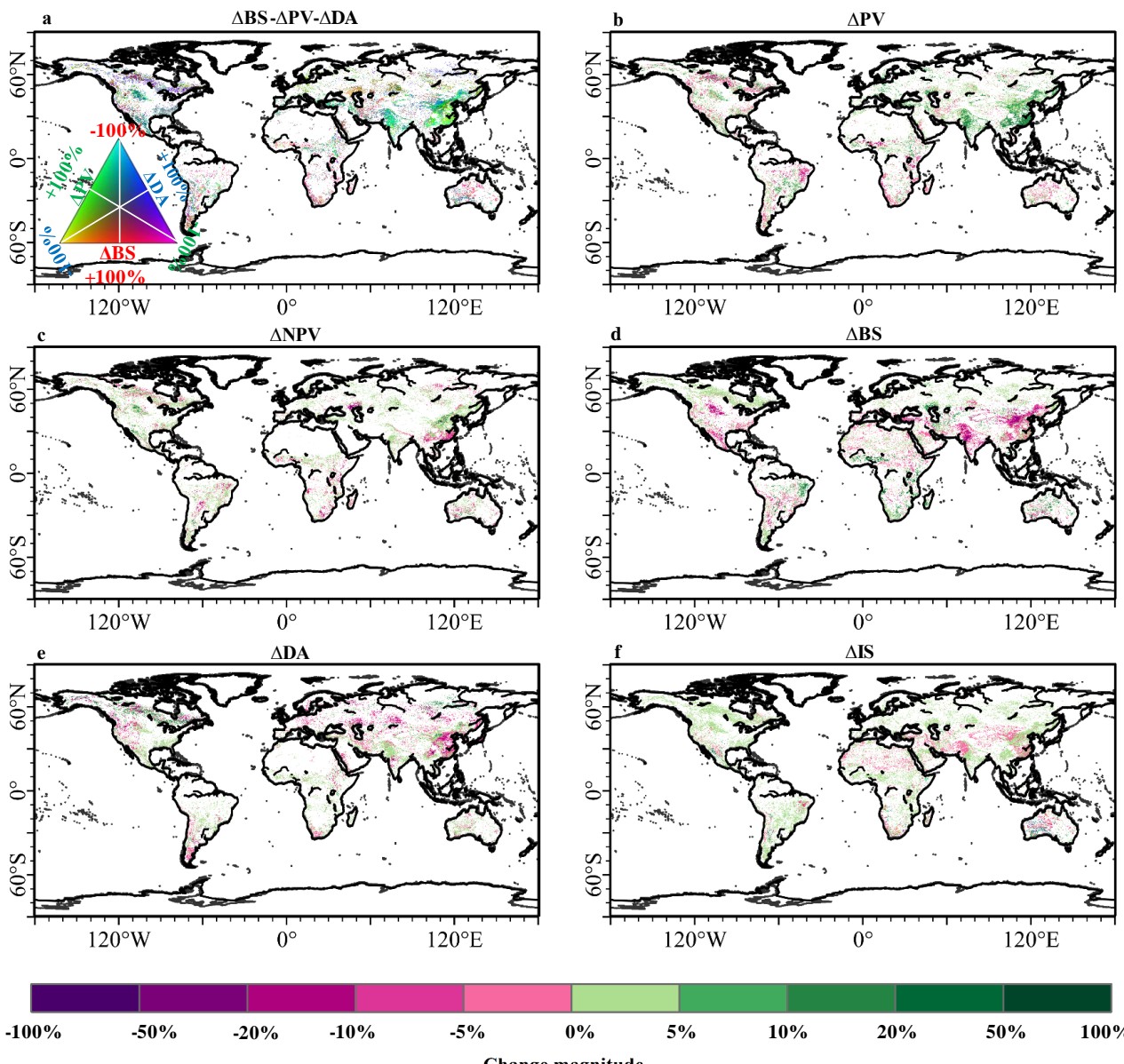


**Figure 7: Globally fractional endmembers dynamics at pixel level. a**, composited RGB image with ΔBS, ΔPV, and ΔDA. **b-f**, the change magnitude (%) in each pixel for estimated endmembers, i.e., ΔPV, ΔNVP, ΔBS, ΔDA, and ΔIS. Pixels showing a statistically significant trend (Seasonal Mann–Kendall test, $P < 0.05$) for either endmember are depicted on the change map.

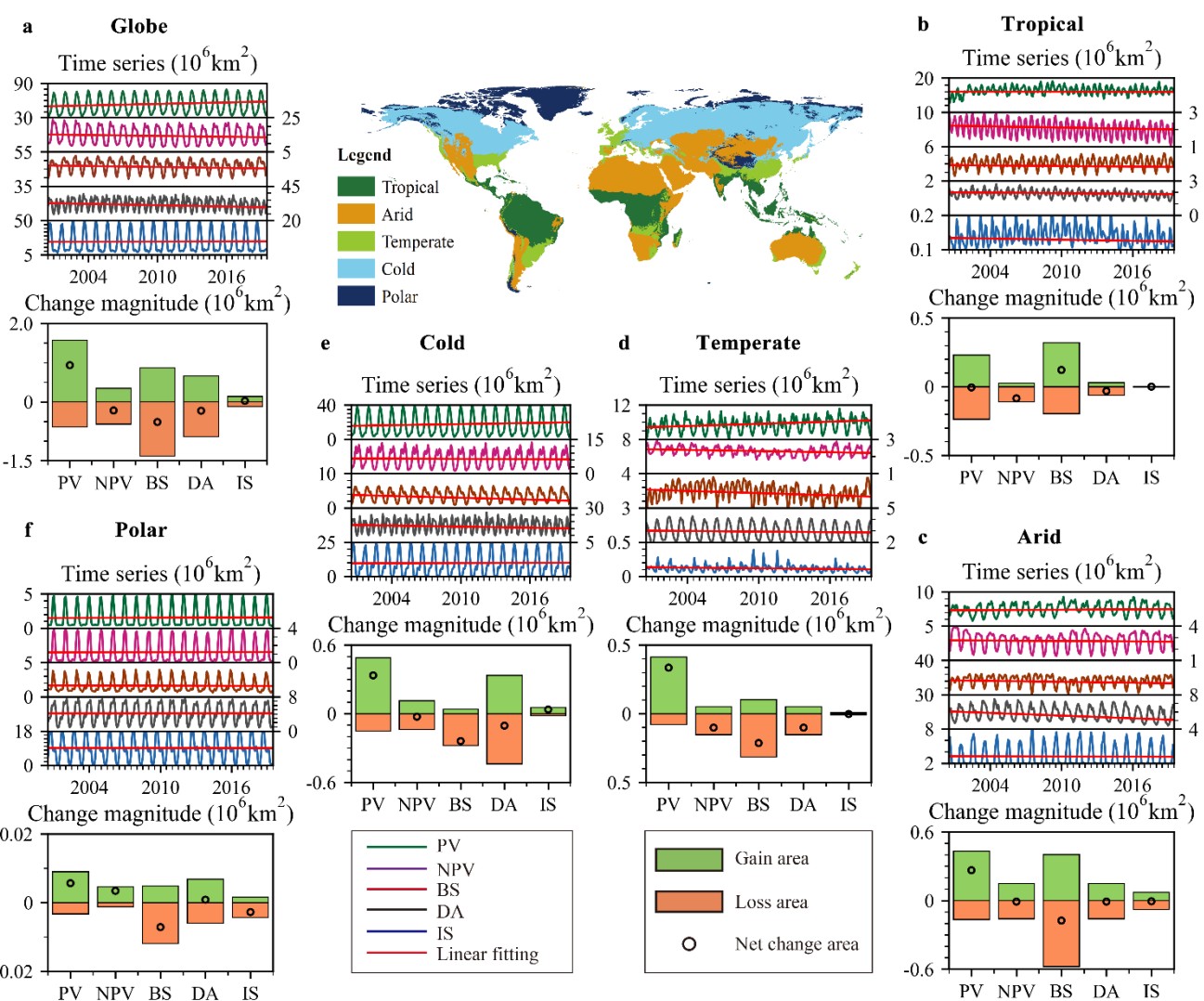

**Figure 8: Global and regional fractional endmembers dynamics.** The middle subgraph is aggregated five Köppen-Geiger
climate classes. **a-f**, the gain area, loss area and net change area for five land surface endmembers in globe (**a**) and five
climate zones, i.e., tropical (**b**), arid (**c**), temperate (**d**), cold (**e**), and polar (**f**).
**4 Discussions**
**4.1 Advances and limitations of estimates of global vegetation and soil fractions**
This paper implements a globally monthly estimates of fractional vegetation-soil nexuses in 2001–2022 via high-accuracy
and time-consuming MESMA algorithm at sub-pixel scale (Roberts et al. 1998), benefited from the GEE platform that can
provide powerful computational processing to realize planetary-scale analysis of geospatial data. We can more
conveniently target the most optimal model from 692 combination models for each MODIS pixel (500 m), thus help to
understand the specific vegetation-soil compositional structures in each pixel or region. Such scheme can improve the
ecologists and managers understanding of multifaceted terrestrial ecosystems for differentiated measures. Moreover, these
fractional endmembers have been proven their potential for application in land use cover classification (Sun et al., 2020),
time-series evolutionary pathways (Sun et al., 2021; Daldegan et al., 2018) and biophysical process modelling (Sun et al.,
2022; Sousa and Small, 2018). This globally comprehensive record of monthly vegetation and soil fractions during the
period 2001–2022 may provide basic data for quantification and modelling of global change, as well as provide an
important foundation for measuring sustainable development goals such as land degradation neutrality (Chasek et al., 2019;
Sun et al., 2019).
Our product can overcome the problem of saturation of NDVI in the regions embodying high coverage vegetation. Such
advance can be supported by previous regional comparison research (Rogan et al. 2002; Sun et al. 2019; Sun et al. 2020).
Additionally, the diversity of information stands as one of the strengths of this dataset, encompassing the five primary
components of the Earth's land surface globally. Moreover, it can be extended to encompass more types through different
levels of clustering. For instance, the DA component has not been emphasized in many datasets, yet current scientific
research underscores the need for increased attention to vegetation shadows (Zeng et al., 2023). Although our DA
component represents various types across different land regions, such as water bodies, shadows, bare rocks, this dataset
may effectively enhance our precise understanding of complex vegetation structures. The NPV component is a vital
element in arid ecosystems and represents a crucial part of vegetation biomass. Our dataset, by finely characterizing NPV,
not only aids in understanding the evolving features of vegetation structure under photosynthetic and non-photosynthetic
interactions (Guerschman et al. 2015), but also contributes to a more accurate quantification of global biomass in arid land
systems (Smith et al. 2019).
Moreover, our product demonstrates good scalability in terms of time and endmember types. These monthly estimates of
fractional vegetation-soil nexuses can be upgraded to multi-timescale (daily, yearly) products to serve different needs, and
thus provide time series of multicomponent information on surface heterogeneous composition and interactive evolution.
Besides, considering the meaningful physical interpretations of endmember fraction values, these endmembers can be
conveniently integrated across different temporal and spatial scales using spatiotemporal fusion methods (Zhang et al.
2018). The temporal and spatial variability of endmembers has always been a significant constraint in obtaining global-
scale vegetation and soil fractions from imagery (Wang et al. 2021). The spatio-temporally adaptive framework employed
helps to increase the representativeness of endmember selection, and MESMA also considers the suitability of each
combination of these endmembers within each pixel. However, considering the limitations of computational resources, our
solution on hierarchical clusters of the endmember spectra can improve considerably cost-effective unmixing of long time-
series satellite records over globe under the trade-offs of certain accuracy requirements (Fig. 3). With the assumption of
increased computational power in the future, we believe that utilization of combination models from selected endmember
spectra (35 GV spectra, 40 BS spectra, 25 NPV spectra, 16 DA spectra, and 15 IS spectra) or expanded endmember spectra
may further improve the accuracy and stability of estimates of gradations of five surface vegetation and soil components
at global scale.
However, due to the absence of corresponding reference data for validation, we solely rely on two high-quality land cover
reference datasets for validation. Unfortunately, these datasets do not intricately characterize small-scale shadows and bare
soil within complex vegetation structures. Consequently, this leads to a misconception in the validation, where our DA and
BS are overestimated in low-value areas and vegetation is underestimated in high-value areas (Fig. 3, Fig.S5). Therefore,
in the future, there is a need to further develop high-quality relevant reference data. Considering that MOD44B vegetation
continuous fields product provides a gradation of three surface cover components: percent tree cover, percent non-tree
cover, and percent bare, the dark components (i.e., shadow of vegetation and mountain, water) are not quantified. Therefore,
fractional PV and NPV is overall biased high, especially in areas with PV and NPV less than 0.50 (Fig. 4b; Fig. S6b).
Besides, we also observed a certain degree of underestimation in these three datasets in regions with lower vegetation cover
compared to our data. This is mainly because these datasets focus solely on green vegetation, especially GLASS and GEOV
Fcover (Baret et al. 2013; Jia et al., 2015), and do not accurately estimate non-photosynthetic vegetation in arid regions.
The above comparisons demonstrate our precision advantage in fine extraction of multiple endmembers. Given the
importance of NPV in ecological research, undertaking separate validation and comparisons between PV and NPV
represents a critical foundational effort. While detailed maps of a representative region illustrate the reliability and
advantages of our NPV estimation over other products (Fig. S6), the current lack of equivalent products highlights the need
for ongoing development. Enhancing quantitative comparison efforts will be essential to bolster the feasibility, accuracy
and validity of our NPV product in future studies. We observed higher $RMSE_{sma}$ values in seemingly homogeneous areas
like the Sahara Desert and Arctic regions. However, within these regions, there often exist extremely diverse land cover
types, such as high and low reflectance sands and ice. When selecting endmembers and hierarchical clustering models, we
might not have adequately considered these extreme spectral curves. As a result, these extreme areas exhibit a higher
uncertainty.
**4.2 Implications of global and regional shifts from pairs of two endmembers**
We find greening of Earth characterized by increased photosynthetic vegetation and reduced bare soil exposure, is observed
in temperate and cold countries such as Russia (Fig. 9; Figure S3). This finding is in agreement with the finding of climate-
driven greening trend in Northern Hemisphere (Piao et al. 2006). While the biomass decreases, exhibited as decreased PV
and increased BS (Fig. 9), presented only half of the global climate-driven greening. These findings imply a global trend
towards greening in the context of global warming, as supported by a large number of published studies on global
vegetation change (Chen et al. 2019; Piao et al. 2006; Song et al. 2018). Moreover, the polar zone is hotspot of ice melting
and agrees an accepted fact of accelerated retreat of glaciers and ice under global warming (Hugonnet et al. 2021; Zemp et
al. 2019).
Besides, the overexploitation of resources is one of environmental problems of interest and an important factor in causing
above climate change and disasters. Global overexploitation has led to problems such as vegetation degradation and
intensive utilization of agricultural land. The human overexploitation of forest and grassland induced biomass decrease
present a decrease of PV and increase of BS (Fig. 9; Fig. S7), especially over tropical rainforest of Brazilian Amazon and
South Asian. This finding agrees with deforestation and agriculturalization in these regions provided by previous studies
(Qin et al. 2019; Zeng et al. 2018). Within agricultural area, the agricultural intensification is a human-driven greening
process characterized by increased photosynthetic vegetation and reduced bare soil, this shift mainly occurs in India and
the North and Northeast China Plain (Fig. 9; Fig. S7) (Chen et al., 2019). We also found urbanization-driven biomass
decrease in the global terrestrial ecosystems, especially in China and North America (Fig. 9; Fig. S7), resulted from
occupation of agricultural and ecological lands during urban sprawl (Kuang et al., 2020, 2021; Zhao et al., 2022).
Eco-restoration depicts a process that currently needs urgent attention in our understanding and utilization of resources and
environment. Different from climate-driven greening that presents trends of increasing PV and decreasing BS, the human-
driven afforestation shows positive trends of both PV and NPV, mainly attributed to recent implementing of policies on
the ecological restoration through large number of protective forests planted (Fig. 9; Fig. S7). These afforested regions are
primarily found over China, Europe, North America, supported by previous study on greening world (Chen et al. 2019).
Moreover, Green space construction in urbanized regions has been carried out, integrated with road construction and city
renovation, and generate an increasing of footprint of urban greening, especially in China (Fig. 9; Fig. S7).
This dataset can serve as a baseline for enhancing our comprehension of heterogeneous surface dynamics and modeling
Earth's biophysical processes through a multi-endmember coupling perspective, may significantly advance future research
by serving as a foundational reference for delving deeper into complex land systems. Anticipating its potential applications
across diverse domains such as ecology, climate studies, and urban planning, this dataset emerges as a pivotal resource. Its
multifaceted utility is expected to play a pivotal role in informing environmental management decisions, advancing studies
on ecological shifts, predicting climate trends, and facilitating strategic landscape planning.

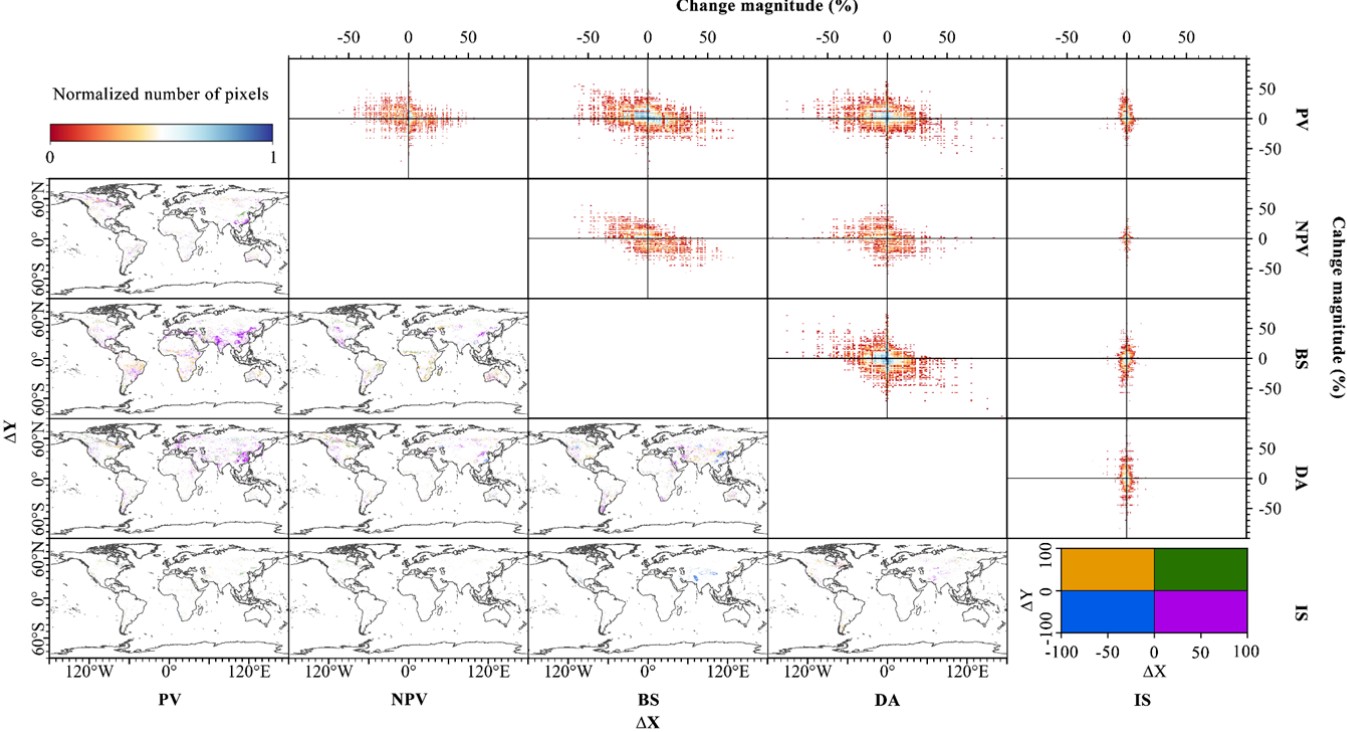


**Figure 9: Characteristics of each pair of two endmembers**. The bottom left corner was global maps of co-location of paired two endmembers. Pixels showing a statistically significant trend (Seasonal Mann–Kendall test, P < 0.05) in both endmembers are depicted on the map. The color of each pixel was displayed in quadrant of ΔX and ΔY, where ΔX and ΔY are horizontal and vertical endmembers, respectively. The top right corner was 2D histogram of change magnitude (%) of paired two endmembers. the x-axes and y-axes were represented by ΔX and ΔY, respectively. These 2D histogram plots were created with bin size of 1% for both axes. The colour bar was normalized number of pixels in each bin on a log scale.

## 5  Data availability

The data about fractional five surface vegetation and soil components can be exported from GEE platform via provided codes or are available on Science Data Bank (https://doi.org/10.57760/sciencedb.13287, Sun and Sun, 2023). The first dataset includes five fractions from 2001-2011, another includes five fractions from 2012-2022. The file is a compressed month-by-month GeoTIFF data for each year, according to the grid of longitude 60° and Latitude 50°. Since the dataset for each year includes 216 files, named as "SMA_year_(month-1)_gridid.tif", like "SMA_2001_0_0.tif". The public datasets have been listed in the Methods.

## 6 Code availability

The GEE codes for the MESMA and seasonal Mann-Kendall test will be available at GitHub (https://github.com/qiangsunpingzh/GEE_mesma) or other platforms upon publication; Common code for generating figures is available at https://matplotlib.org/.

## 7  Conclusions

In this paper, to provide locally detailed socio-ecological knowledge about globally multifaceted changes in fractional vegetation-soil nexuses under climate change and anthropogenic impacts, we estimated monthly vegetation and soil fractions in 2001–2022 that provide multi-component information on surface heterogeneous composition based on a spatio-temporally adaptive spectral mixture analysis framework. This product of monthly vegetation and soil fractions from 692 combination models can provide an accurate estimate of surface heterogeneous composition, better than previous vegetation index and vegetation continuous fields product, as well as traditional fully constrained linear spectral mixture models. This solution can both improve considerably cost-effective unmixing of long time-series satellite records over globe and meet the accuracy requirements. Based on these estimates of vegetation and soil fractions, we find a greening trend of Earth, as indicated by a increase of the total area of PV, which represents a +1.88% change relative to 2001 green vegetation. This greening trend can be found all climatic zones other than the tropics. In addition to the trends in the greening reported by other study, we also found that the increase in PV was accompanied by a decreasing trend in BS, DA and NPV in most regions. And there is a trend of simultaneous increase in PV and NPV in central and southwest China during afforestation activities. Therefore, a combination between interactive changes of vegetation and soil fractions can be adopted as a valuable measurement of climate change and anthropogenic impacts.

## Author contributions

Q.S., D.S., P.Z., and H.L. designed the study. Q.S. and P.Z. performed the analysis with support from D.S., H.L., J.H., S.L., and S.Y. Q.S., P.Z., and D.S. drafted the paper. Q.S., P.Z., X.J., M.S., F.L., W.D., S.M., A.L., Y.Z., and H.L. collected data and prepared figures. All authors contributed to interpretation of the results and discussions as well as manuscript editing.

## Competing interests

The contact author has declared that none of the authors has any competing interests.

## Acknowledgements

We thank Bruce W. Pengra for providing the GLCVRD reference dataset.

## Financial support

Funding for this work was provided by National Key R&D Program of China (grant no. 2023YFB3907604) and the National Natural Science Foundation of China (grant nos. 42001234, 42071252).

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
