# Peer review of "A global estimate of monthly vegetation and soil fractions from"

_Earth System Science Data, 2023_

## Author Comment (AC1)

**Responses and Revisions to Comments**

**Dear reviewer**

Thanks for your reviewing and valuable comments of our manuscript entitled "***A global estimate of monthly vegetation and soil fractions from spatio-temporally adaptive spectral mixture analysis during 2001–2022***". We also appreciate you for providing insightful feedback and comments to strengthen our manuscript.

We have revised our manuscript with considering each detailed suggestion that you have graciously provided. These major revisions include:

1 More comprehensive description of the validation and comparison processes.

2 Additional land cover and land use reference data and fractional vegetation cover data for validation and comparison.

3 A more accurate representation.

Besides these revisions, all authors checked the manuscript carefully and several minor revisions have been done to finalize the manuscript.

The following is a point-by-point response to the questions and comments delivered in your letter. For your convenience, revisions made by the authors have been highlighted in red color in the both response and revised manuscript, which could be easily checked. We hope that our revisions and responses can satisfactorily address all the issues and concerns.

*Spatially-explicit monitoring of vegetation and soil fractions is critical for understanding terrestrial ecological processes. There are thus many global vegetation fractional cover products including ENVISAT, CYCLOPES, GEOV, MuSyQ, GLASS, and CGLS. The authors provided a unified monthly fractional vegetation-soil nexuses product during 2001-2022 with MESMA. As a user, we would like to rely on more precise satellite-derived fraction data. Therefore, I pay more attention to the credibility of the new fraction products presented in this paper. This paper merely presented the evaluation of the estimates of vegetation and soil fractions datasets in Section 3.1 and Section 4.1, respectively. However, this validation is inadequate. It's difficult to persuade me that the presented product is superior to existing fraction products. Thus, further validations are required.*

We value your thoughtful recommendations regarding the validation of vegetation and soil fractions. Acknowledging the crucial role of data reliability in accurately depicting land surface processes, our manuscript incorporates CLCVRD data along with other land cover and land use reference data for validating the estimated vegetation and soil fractions in Sections 3.1, and provides additional fractional vegetation cover data for comparison in Section 3.2.

First, to enhance the validation process for the showcased product, we have enhanced the procedural description for estimating fractional vegetation-soil compared to GLCVRD, ensuring product reliability (page 11, line 236-251).

"Moreover, due to challenges in conducting fraction estimation validation through field surveys, we employ reference data obtained from high spatial resolution images as validation set. We thus select for two sets of reference data that their land cover classification systems are closely related to our five endmembers.

...

Firstly, we filter the estimated fractions based on the corresponding year and month obtained from the reference data. Simultaneously, aligning the interpretations of land cover types with our endmembers, we pair them accordingly, that is, tree and other vegetation represent PV and NPV, barren stands for BS, water and shadow correspond to DA, and ice & snow denote IS. Subsequently, we reclassify these paired land cover types and calculated their percentage within 5×5 km blocks, in which we exclude cloud coverage (named no data). Additionally, utilizing these cloud-free pixels in each block, we compute the mean of fractional values for each endmember, and then compare these estimated fractions with the measured percentage of paired the reclassified land cover types to validate the reliability of our product (Fig. S4)."

We further authenticate our product through incorporating comprehensive global land cover and land use reference data, which were obtained from the Geo-Wiki crowdsourcing platform across four campaigns[1] (page 12, line 265-276).

"Besides, we also authenticate our product through incorporating comprehensive global land cover and land use reference data (Fritz et al. 2017), which were obtained from the Geo-Wiki crowdsourcing platform across four campaigns: Human impact, wilderness, reference and disagreement. Over 150000 samples of land cover and land use were acquired in this reference data. To effectively validate our product, we need to filter the reference data, considering aspects such as data acquisition time, measurement methods, and credibility. We select first three campaigns, which have a good match with MODIS pixels (size 1×1km) and were observed during 2001 to 2022, and then select 1038 high feasibility reference data through the confidence information of land cover estimates and the status of use of high spatial resolution imagery provided by the metadata. Finally, Similarly to the procedural description used for fractional vegetation-soil compared to GLCVRD, we reclassified ten classes of this dataset into our four groups of endmembers, including (1) tree cover, shrub cover, herbaceous vegetation/grassland, cultivated and managed, and mosaic of cultivated and managed/natural vegetation to PV and NPV; (2) flooded/wetland and open water to DA; (3) urban and barren to BS; (4) snow and ice to IS. This involved

[1] Fritz, S., See, L., Perger, C., McCallum, I., Schill, C., Schepaschenko, D., Duerauer, M., Karner, M., Dresel, C.,r Laso-Bayas, J. C., Lesiv, M., Moorthy, I., Salk, C. F., Danylo, O., Sturn, T., Albrecht, F., You, L., Kraxner F., Obersteiner, M.: A global dataset of crowdsourced land cover and land use reference data. Scientific data, 4, 1-8, https://doi.org/10.1038/sdata.2017.75, 2017.

comparing the measured percent of land cover with the mean of endmember fractions within the corresponding 1x1km pixels."

[Figure]

**Fig. S5 Evaluation of global fractional endmember estimates based on land cover reference data.** a, the location of high-feasibility land cover reference data. b-d, Scatter plots of PV+NPV, BS, DA, IS fractions against land cover reference data

Moreover, in addition to the original two datasets, we included two additional datasets recommended by you for conducting relevant comparisons. Finally, we strengthened the comparisons between our generated data and four existing datasets: NDVI, MOD44B Vegetation Continuous Fields product, GLASS fractional vegetation cover dataset, and GEOV Fcover dataset in Section 2.4 Comparisons and uncertainties analysis (page 12-13, line 278-293), and present results in Section 3.2 Compared with other datasets and traditional spectral mixture analysis model. Such comparisons include bi-dimensional histogram of fractional endmembers and other dataset (Fig. 4) and detailed figures (Fig. S6).

"To verify the consistency and merits of our dataset against existing ones, we conducted comparisons with four distinct pre-existing datasets: NDVI, MOD44B Vegetation Continuous Fields product, GLASS fractional vegetation cover dataset, and GEOV Fcover dataset. NDVI is derived from monthly synthesized MCD43A4 images. Both mean values of NDVI and our estimated fractional PV across all years and months are considered for comparison. The MOD44B Vegetation Continuous Fields product provides annual information about the percent tree cover, percent non-tree cover, and percent non-vegetated within each 250-meter pixel globally (DiMiceli et al., 2015). Consequently, we compare vegetation cover proportions—sum of percent tree cover and percent non-tree cover—to the sum of fractional PV and NPV. To align spatial and temporal resolutions, we aggregated the sum of percent tree cover and percent non-tree cover to a 500-meter scale. Simultaneously, we computed monthly Fractional PV and NPV as annual averages. The GLASS fractional vegetation cover dataset, offering an 8-day temporal frequency and dual spatial resolutions of 0.05° and 500 meters, was generated using a machine learning approach correlating MODIS reflectance with fractional vegetation cover (Jia et al., 2015). In our study, the 500-meter GLASS data was utilized to validate our estimated fractions. We computed annual averages from all the CLASS fractional vegetation cover data within a year and compared it with the annual averages of Fractional PV and NPV. GEOV FCover is a 10-day product estimated through the neural network using visible, near-infrared and shortwave infrared at 1km resolution (Baret et al. 2013). We aggregate our product to a 1km spatial resolution, and compare their annual averages with the annual averages of GEOV FCover."

[Figure]

**Figure 4: Comparisons with other datasets and traditional spectral mixture analysis models. a, b, c, d** the bi-dimensional histogram of fractional endmembers and other dataset with bin size of 2%, including fractional PV against NDVI (a), fractional PV and NPV against fractional tree and non-tree vegetation of MOD44B vegetation continuous fields product (b), fractional PV and NPV against GLASS fractional vegetation cover product (c), fractional PV and NPV against fractional vegetation cover of GEOV Fcover product; **e, f**, the boxplot and violin plot for average of monthly $RMSE_{sma}$ for two fixed endmember spectral curves using fully constrained linear spectral mixture models, including (e) average of all spectral spectra for each endmember and (f) existing spectral spectra from Small and Sousa (2019).

[Figure]

**Fig. S6 The detailed graphs for comparing different datasets**. a, b, and c represent comparisons of vegetation abundance products in different scenarios, specifically, regions with low vegetation cover in arid areas, high vegetation cover in tropical rainforests, and transitional zones from low to high values. The compared products include our produced PV and NPV, MOD44B, GLASS, and GEOV.

*MESMA needs a large library of endmembers representative of each ground component. In Section 2.2.1, the paper depicted the selection of endmembers. I suggest that supplement some RGB images of these selected pure pixels.*

Thank you very much for your valuable suggestions, we've chosen several typical images representing selected pure pixel of each endmember. These images have been used as part of the supporting material (supplement) to underpin the reliability of the data.

[Figure]

**Figure S2: Typical images representing selected pure pixel of each endmember**. a-o are tropical rainforest, temperate forest, cropland, grasslands, temperate deciduous forest in winter, crop residues, shrubs in dryland, moving sands, sand dunes, bare ground, moving sands. waters, bare rock, polar glaciers, and alpine glaciers.

*Please clarify the reason that the mean error and mean absolute error were used as accuracy metrics.*

The mean error (ME) and mean absolute error (MAE) are commonly used accuracy metrics in various fields, especially in statistics and machine learning, for assessing the performance of predictive models.

Mean Error (ME): ME measures the average of all errors in a dataset where errors are the differences between predicted and actual values. ME helps in understanding the overall bias of the model. Mean Absolute Error (MAE): MAE is the average of the

absolute differences between predicted and actual values. It measures the average magnitude of errors without considering their direction. It's easier to interpret as it provides a straightforward understanding of the average prediction error.

Both ME and MAE are useful metrics to evaluate the accuracy of predictive models, the use of ME and MAE depends on the specific context of the problem and what kind of error measurement is more important for the analysis[2].

We thus improve the descriptions of these four accuracy metrics (page 11, line 253-258).

"ME measures the average of all errors in the dataset where errors are the differences between predicted and actual values, MAE calculates the average of the absolute differences between predicted and actual values, RMSE provides a measure of prediction error, whereas $R^2$ offers insight into the amount of variability in the dependent variable that the model explains. These metrics provide a more comprehensive assessment of the model's accuracy, helping to understand different facets of its performance, such as bias, variability, and overall predictive power (James et al., 2013)."

*Please provide additional information on the source of GLCVRD, such as GLCVRD (Bruce Pengra et al., 2015).*

Thanks for your valuable suggestions, we have provided additional information on the source of GLCVRD (Olofsson et al. 2012; Pengra et al. 2015; Stehman et al. 2012)[3]. (page 11, line 240)

"Global Land Cover Validation Reference Dataset (GLCVRD) is provided with a 2m reference dataset from very high resolution commercial remote sensing data within 5 × 5 km blocks from 2003 to 2012 (Olofsson et al. 2012; Pengra et al. 2015; Stehman et al. 2012)"

*Could you please provide an explanation for the red dot appearing at the top of Figure 3?*

The overlaid red dots in Figure 3a were spatial distribution of the 5 × 5 km validation blocks of GLCVRD reference dataset.

We have included this explanation in page 15, line 338-339:
* * *
[2] James, G., Witten, D., Hastie, T., Tibshirani, R.: An introduction to statistical learning. New York: springer, https://doi.org/10.1007/978-1-0716-1418-1. 2013

[3] Olofsson, P., Stehman, S.V., Woodcock, C.E., Sulla-Menashe, D., Sibley, A.M., Newell, J.D., Friedl, M.A., Herold, M.: A global land-cover validation data set, part I: Fundamental design principles. Int. J. Remote Sens., 33, 5768-5788, https://doi.org/10.1080/01431161.2012.674230,2012.

Pengra, B., Long, J., Dahal, D., Stehman, S.V., Loveland, T.R.: A global reference database from very high resolution commercial satellite data and methodology for application to Landsat derived 30 m continuous field tree cover data. Remote Sens. Environ., 165, 234-248, https://doi.org/10.1016/j.rse.2015.01.018, 2015.

Stehman, S.V., Olofsson, P., Woodcock, C.E., Herold, M., Friedl, M.A.: A global land-cover validation data set, II: Augmenting a stratified sampling design to estimate accuracy by region and land-cover class. Int. J. Remote Sens., 33, 6975-6993, https://doi.org/10.1080/01431161.2012.695092, 2012.

"The overlaid red dots were spatial distribution of the 5 × 5 km validation blocks of GLCVRD reference dataset (n=500)"

*Could you please provide a detailed procedure for fractional vegetation-soil estimates compared to GLCVRD?*

Thanks for your valuable suggestions, to enhance the validation process for the showcased product, we have enhanced the procedural description for estimating fractional vegetation-soil compared to GLCVRD through detailed textual descriptions and supporting figures (page 11, line 244-251).

"Firstly, we filtered the estimated fractions based on the corresponding year and month obtained from the reference data. Simultaneously, aligning the interpretations of land cover types with our endmembers, we paired them accordingly, that is, tree and other vegetation represent PV and NPV, barren stands for BS, water and shadow correspond to DA, and ice & snow denote IS. Subsequently, we reclassified these paired land cover types and calculated their percentage within 5×5 km blocks, in which we excluded cloud coverage (named no data). Additionally, utilizing these cloud-free pixels in each block, we computed the mean of fractional values for each endmember, and then compared these estimated fractions with the measured percentage of paired the reclassified land cover types to validate the reliability of our product (Fig. S4)."

[Figure]

**Figure S4: Procedural description for estimating fractional vegetation-soil compared to GLCVRD.**

*Line 332 "estimates vegetation and soil fractions": estimates of vegetation and soil fractions*

Thank you very much for your careful review of our manuscript, we have corrected these errors and have also carefully examined and revised the manuscript to ensure the accuracy of the presentation.

---

## Author Comment (AC2)

**Revisions and Responses to Comments**

**Dear reviewer**

Thanks for your reviewing and valuable comments of our manuscript entitled "*A global estimate of monthly vegetation and soil fractions from spatio-temporally adaptive spectral mixture analysis during 2001–2022*". We also appreciate you for providing insightful feedback and comments to strengthen our manuscript.

We have revised our manuscript with considering each detailed suggestion that you and reviewers have graciously provided. These major revisions include:

1 Enhanced the introduction and discussion regarding the significance of this dataset.

2 Augmented experiments with more data for validation and comparison to prove the data reliability.

3 Conducted in-depth analysis and discussion on the validations and comparisons.

4 Adjusted the structure of the article.

5 Improved the expression of the article.

Besides these revisions, all authors checked the manuscript carefully and several minor revisions have been done to finalize the manuscript.

The following is a point-by-point response to the questions and comments delivered in your letter. For your convenience, revisions made by the authors have been highlighted in red color in the both response and revised manuscript, which could be easily checked. We hope that our revisions and responses can satisfactorily address all the issues and concerns.

*Monthly vegetation and soil fractions products are important to understand the global landcover change and evaluate the impacts of climate change and human activities on the terrestrial ecosystem. The topic is interesting and the MESMA algorithm may be a practical method for decoupling the mixed pixels. I acknowledge the potential of the algorithm; however, I have some major concerns regarding the dataset itself, including its significance, validation methods, as well as the overall organization of the article. Given that the ESSD journal focuses more on the dataset, I am unable to provide a positive evaluation.*

Thank you very much for your feedback on our manuscript. We understand your concerns regarding the significance and reliability of the vegetation and soil fractions products presented in our manuscript, possibly due to our insufficient description of advantages and validation. We believe our dataset holds certain advantages in terms

of its significance and reliability. Our response primarily encompasses the following points:

[Significance] (1) Our dataset surpasses traditional vegetation index and fractional vegetation cover datasets by offering a comprehensive view of various surface elements including vegetation, photosynthetic vegetation, bare soil, dark material, ice and snow. This expanded coverage not only enhances our comprehension of heterogeneous surface dynamics but also presents distinct advantages for modeling Earth's biophysical processes. Both our team and researchers have validated these advantages[1]. Consequently, we firmly believe in the practical significance of our dataset. (2) Unlike high-resolution fractional vegetation cover data typically constrained by temporal resolution limits, our dataset presents a unique perspective on surface evolution across extended periods (>20 years) and at high frequencies. Meanwhile, Currently, the academic community extensively utilizes high temporal resolution images like MODIS and AVHRR to analyze global surface changes and biophysical processes [2], This suggests our 500-meter product stands out as exceptionally valuable globally. (3) Moreover, owing to the intrinsic significance of these surface endmembers, achieving high spatial and temporal resolution abundance data through spatiotemporal fusion[3] becomes notably easier compared to traditional spectral or spectral index approaches. This attribute underscores the dataset's ability to capture intricate surface dynamics with enhanced precision.

We have included a review about vegetation index and fractional vegetation cover in Section of Introduction (page 3, line 72-87) and discussed advantage of our product in Section 4.1 (page 24-25, line 455-469).

[revised manuscript text omitted]
. Methodologically, we've maximized our model optimizations within the constraints of current computational capabilities. This involved leveraging endmember curves from five distinct endmember types, totaling 15 subclasses. Through meticulous optimization, we obtained superior unmixed results from a pool of 692 models. Notably, the model's uncertainty has been thoroughly elucidated in Section 3.3. While our current computational resources set the boundaries, our aim is to further refine and amplify these models in the future. This includes expanding the spectrum of endmember types and increasing the count of endmember spectral curves using the provided code, provided that computational resources allow.

We thus discussed the reliability of our product in Section 4.1 Advances and limitations of estimates of global vegetation and soil fractions

*1. For significance and usefulness, the product is derived using the MODIS data with a resolution of 500m. Actually, there are a lot of available global or regional higher-resolution land cover products and FVC products. As a user, why do I have to select your product for the analysis? I think using higher-resolution products can acquire more reasonable results. There is not enough evidence in the article that convince me of the indispensability of your dataset at this stage. More content indicating the importance should be included in the Introduction.*

Thank you very much for your valuable suggestions. Although there are numerous high spatial resolution land cover and fractional vegetation cover products available, we believe our dataset holds certain advantages for conducting global-scale analyses. This is primarily showcased in the following aspects:

**Unique Comprehensive Insights**: We highlight the distinctiveness of the dataset in providing a comprehensive view that encompasses fractional multiple surface features (including vegetation, photosynthetic vegetation, bare soil, dark material, ice and snow) beyond traditional land cover or fractional vegetation cover products. This breadth of information can offer a holistic understanding of surface dynamics globally.

**Analytical Advantages**: Our data exhibits favorable advantages in extended periods and at high frequencies, which aids in discovering finer processes. Considering the physical significance of the endmembers, it contributes to our dataset's ability to conduct clear downscaling and upscaling at the spatiotemporal scale[5]. Moreover, the scalability of the MESMA assists us in expanding the dataset's accuracy in the future by broadening endmember types and spectral curves. This dataset can serve as a baseline or reference for enhancing our comprehension of heterogeneous surface dynamics and modeling Earth's biophysical processes in the future studies, its potential applications in various fields such as ecology, climate studies, or urban planning.

By incorporating these into the Introduction and Discussions, we can provide a more compelling argument for the indispensability and significance of our dataset, effectively addressing the concerns of potential users seeking higher-resolution products for analysis.

Introduction (page 3, line 75-87)

"In recent years, there have been significant advancements in fractional vegetation cover within the fields of remote sensing and environmental science. This progress has led to the development of various products at multiple resolutions, such as Long-term global land surface satellite (GLASS), GEOV Fcover, Multi-source data Synergized Quantitative remote sensing production system (MuSyQ) fractional vegetation cover (Baret et al. 2013; Jia et al., 2015; Mu et al., 2017; Zhao et al., 2023). These products primarily integrate and utilize data from different spectral bands and sensors, employing methods including machine learning and radiative transfer model. However, these data primarily focus on green vegetation, posing significant limitations in capturing information regarding non-photosynthetic vegetation and bare soil. This constraint also restricts the applicability of this data in arid regions."

[5] Small, C., Milesi, C.: Multi-scale standardized spectral mixture models. Remote Sens. Environ., 136, 442-454, https://doi.org/10.1016/j.rse.2013.05.024, 2013.

Zhang, Y., Foody, G. M., Ling, F., Li, X., Ge, Y., Du, Y., & Atkinson, P. M.: Spatial-temporal fraction map fusion with multi-scale remotely sensed images. Remote Sensing of Environment, 213, 162-181, 2018.

Discussions (page 24, line 443-454):

"Our product can overcome the problem of saturation of NDVI in the regions embodying high coverage vegetation. Such advance can be supported by previous regional comparison research (Rogan et al. 2002; Sun et al. 2019; Sun et al. 2020). Additionally, the diversity of information stands as one of the strengths of this dataset, encompassing the five primary components of the Earth's land surface globally. Moreover, it can be extended to encompass more types through different levels of clustering. For instance, the DA component has not been emphasized in many datasets, yet current scientific research underscores the need for increased attention to vegetation shadows (Zeng et al., 2023). Although our DA component represents various types across different land regions, such as water bodies, shadows, bare rocks, this dataset may effectively enhance our precise understanding of complex vegetation structures. The NPV component is a vital element in arid ecosystems and represents a crucial part of vegetation biomass. Our dataset, by finely characterizing NPV, not only aids in understanding the evolving features of vegetation structure under photosynthetic and non-photosynthetic interactions (Guerschman et al. 2015), but also contributes to a more accurate quantification of global biomass in arid land systems (Smith et al. 2019)."

*2. The current classification system comprises only five types. While I understand that the algorithm proposed by the authors can effectively decompose mixed pixels, I am not sure if it is enough for the practical analysis only based on these categories.*

We appreciate your concerns about the choice of five endmember types. A framework encompassing substrate, vegetation, dark, and ice/snow has been proposed and globally validated for both Landsat and MODIS. This framework ensures consistent comparison of estimated fractions across diverse climate patterns and land cover types[6]. To enrich the diversity of surface elements on a global scale, we also expanded the variety of endmember types based on endmember types commonly used in drylands[7]. Therefore, our selection of five endmembers is both feasible and necessary.

We added descriptions regarding the determination of endmember types (page 6, line 163-171).

"Recent studies have proposed various compositional endmember frameworks in different application contexts. For example, a framework including substrate, vegetation, dark and ice/snow was proposed and verified globally for both Landsat
* * *
[6] Sousa, D., Small, C.: Globally standardized MODIS spectral mixture models. Remote Sens Lett, 10, 1018-1027, https://doi.org/10.1080/2150704X.2019.1634299, 2019

[7] Guerschman, J. P., Scarth, P. F., McVicar, T. R., Renzullo, L. J., Malthus, T. J., Stewart, J. B., Trevithick, R.: Assessing the effects of site heterogeneity and soil properties when unmixing photosynthetic vegetation, non-photosynthetic vegetation and bare soil fractions from Landsat and MODIS data. Remote Sens. Environ., 161, 12-26, https://doi.org/10.1016/j.rse.2015.01.021, 2015

and MODIS to allow estimated fractions, this framework ensures consistent comparison of estimated fractions across diverse climate patterns and land cover types (Small and Milesi 2013; Sousa and Small 2019). Another framework includes photosynthetic vegetation, non-photosynthetic vegetation, soil, and shade (Roberts et al. 1993), this framework was widely adopted for presentation surface structure worldwide, particularly in tropical rainforest and dryland ecosystems (Guerschman et al., 2015). These elements can characterize the fundamental composition of the Earth surface. Thus, we embody five endmembers to represent surface units, these five endmembers include photosynthetic vegetation (PV), non-photosynthetic vegetation (NPV), bare soil (BS), dark (DA), ice/snow (IS)"

We also discussed the expandability of endmember types of MESMA models (page 24-25, line 461-469).

"The spatio-temporally adaptive framework employed helps to increase the representativeness of endmember selection, and MESMA also considers the suitability of each combination of these endmembers within each pixel. However, considering the limitations of computational resources, our solution on hierarchical clusters of the endmember spectra can improve considerably cost-effective unmixing of long time-series satellite records over globe under the neglect of certain accuracy requirements (Fig. 3). With the assumption of increased computational power in the future, we believe that utilization of combination models from selected endmember spectra (35 GV spectra, 40 BS spectra, 25 NPV spectra, 16 DA spectra, and 15 IS spectra) or expanded endmember spectra may further improve the accuracy and stability of estimates of gradations of five surface vegetation and soil components at global scale."

*3. The superiority of your dataset is only proved by the comparison with the NDVI and MOD44B product and I think is not adequate for the validation and drawing a reasonable conclusion that your model is more accurate.*

We appreciate your concerns about comparison and validation. Although our dataset's comparison with NDVI and MOD44B products is informative, additional comparison and validation measures would indeed fortify the claim of superior accuracy. We thus explore further validation to bolster the credibility of our model.

First, we further authenticate our product through incorporating comprehensive global land cover and land use reference data, which were obtained from the Geo-Wiki crowdsourcing platform across four campaigns (page 12, line 265-276).

"Besides, we also authenticate our product through incorporating comprehensive global land cover and land use reference data (Fritz et al. 2017)[8], which were obtained

[8] Fritz, S., See, L., Perger, C., McCallum, I., Schill, C., Schepaschenko, D., Dueerauer, M., Karner, M., Dresel, C.,r Laso-Bayas, J. C., Lesiv, M., Moorthy, I., Salk, C. F., Danylo, O., Sturn, T., Albrecht, F., You, L., Kraxner F., Obersteiner, M.: A global dataset of crowdsourced land cover and land use reference data. Scientific data, 4,

from the Geo-Wiki crowdsourcing platform across four campaigns: Human impact, wilderness, reference and disagreement. Over 150000 samples of land cover and land use were acquired in this reference data. To effectively validate our product, we need to filter the reference data, considering aspects such as data acquisition time, measurement methods, and credibility. We select first three campaigns, which have a good match with MODIS pixels (size 1×1km) and were observed during 2001 to 2022, and then select 1038 high feasibility reference data through the confidence information of land cover estimates and the status of use of high spatial resolution imagery provided by the metadata. Finally, Similarly to the procedural description used for fractional vegetation-soil compared to GLCVRD, we reclassified ten classes of this dataset into our four groups of endmembers, including (1) tree cover, shrub cover, herbaceous vegetation/grassland, cultivated and managed, and mosaic of cultivated and managed/natural vegetation to PV and NPV; (2) flooded/wetland and open water to DA; (3) urban and barren to BS; (4) snow and ice to IS. This involved comparing the measured percent of land cover with the mean of endmember fractions within the corresponding 1x1km pixels."

1-8, https://doi.org/10.1038/sdata.2017.75, 2017.

[Figure]

**Fig. S5 Evaluation of global fractional endmember estimates based on land cover reference data.** a, the location of high-feasibility land cover reference data. b-d, Scatter plots of PV+NPV, BS, DA, IS fractions against land cover reference data

Second, we present further comparison with other fractional vegetation cover dataset (page 12-13, line 277-293) using scatter plot (Fig. 4) and detailed images (Fig. S6).

"To verify the consistency and merits of our dataset against existing ones, we conducted comparisons with four distinct pre-existing datasets: NDVI, MOD44B Vegetation Continuous Fields product, GLASS FVC dataset, and GEOV Fcover dataset. NDVI is derived from monthly synthesized MCD43A4 images. Both mean values of NDVI and our estimated fractional PV across all years and months are considered for comparison. The MOD44B Vegetation Continuous Fields product provides annual information about the percent tree cover, percent non-tree cover, and percent non-vegetated within each 250-meter pixel globally (DiMiceli et al., 2015). Consequently, we compare vegetation cover proportions—sum of percent tree cover and percent non-tree cover—to the sum of fractional PV and NPV. To align spatial and temporal resolutions, we aggregated the sum of percent tree cover and percent non-tree cover to a 500-meter scale. Simultaneously, we computed monthly Fractional PV and NPV as annual averages. The GLASS fractional vegetation cover dataset, offering an 8-day temporal frequency and dual spatial resolutions of 0.05° and 500 meters, was generated using a machine learning approach correlating MODIS reflectance with fractional vegetation cover (Jia et al., 2015). In our study, the 500-meter GLASS data was utilized to validate our estimated fractions. We computed annual averages from all the CLASS fractional vegetation cover data within a year and compared it with the annual averages of Fractional PV. GEOV FCover is a 10-day product estimated through the neural network using visible, near-infrared and shortwave infrared at 1km resolution (Baret et al. 2013). We aggregate our product to a 1km spatial resolution, and compare their annual averages with the annual averages of GEOV FCover."

[Figure]

**Figure 4: Comparisons with other datasets and traditional spectral mixture analysis models. a, b, c, d** the bi-dimensional histogram of fractional endmembers

and other dataset with bin size of 2%, including fractional PV against NDVI (a), fractional PV and NPV against fractional tree and non-tree vegetation of MOD44B vegetation continuous fields product (b), fractional PV and NPV against GLASS fractional vegetation cover product (c), fractional PV and NPV against fractional vegetation cover of GEOV Fcover product; **e**, **f**, the boxplot and violin plot for average of monthly $RMSE_{sma}$ for two fixed endmember spectral curves using fully constrained linear spectral mixture models, including (e) average of all spectral spectra for each endmember and (f) existing spectral spectra from Small and Sousa (2019).

[Figure]

**Fig. S6 The detailed graphs for comparing different datasets**. a, b, and c represent comparisons of vegetation abundance products in different scenarios, specifically, regions with low vegetation cover in arid areas, high vegetation cover in tropical rainforests, and transitional zones from low to high values. The compared products include our produced PV and NPV, MOD44B, GLASS, and GEOV.

*4. I think the accuracy of the algorithm shown in Fig.3e-h is not promising. For the BS/DA/IS type, when the reference values are low, why does the algorithm show a pronounced overestimation? I think the authors should explain it in the Discussion. Comparison with other methods or products is necessary here to prove the superiority of your dataset. Additionally, high R-squared values for the IS type are likely attributable to the data being excessively scattered along the X-axis and more samples should be included for this type.*

Your insights regarding the algorithm's accuracy in Fig. 3e-h are valuable. This is resulted from the fact that our estimated DA/BS less than 0.2 is presented as 0 in the reference data, because the interpreted reference dataset of high-spatial satellite observations ignored the shadows of the vegetation and bare soil within tree. In blocks with a DA/BS greater than 0.2, the estimated fractions and measured fractions present better consistency, in which the shadows and soil are well measured by GLCVRD.

We have revised that in the Discussions (page 25, line 470-474)

"However, due to the absence of corresponding reference data for validation, we solely rely on two high-quality land cover reference datasets for validation. Unfortunately, these datasets do not intricately characterize small-scale shadows and bare soil within complex vegetation structures. Consequently, this leads to a misconception in the validation, where our DA and BS are overestimated in low-value areas and vegetation is underestimated in high-value areas (Fig. 3, Fig.S5). Therefore, in the future, there is a need to further develop high-quality relevant reference data."

The IS sample count aligns with other endmembers. However, numerous samples lack snow cover, resulting in our predictions and measurements scattering around the (0,0) coordinate. This condition contributes to a higher $R^2$ for IS. To bolster the reliability of predictive accuracy for these endmember fractions, we've introduced an additional 1083 new samples from incorporating comprehensive global land cover and land use reference data, which were obtained from the Geo-Wiki crowdsourcing platform across four campaigns (page 12, line 265-276).

"Besides, we also authenticate our product through incorporating comprehensive global land cover and land use reference data (Fritz et al. 2017), which were obtained from the Geo-Wiki crowdsourcing platform across four campaigns: Human impact, wilderness, reference and disagreement. Over 150000 samples of land cover and land use were acquired in this reference data. To effectively validate our product, we need to filter the reference data, considering aspects such as data acquisition time,

measurement methods, and credibility. We select first three campaigns, which have a good match with MODIS pixels (size 1×1km) and were observed during 2001 to 2022. High feasibility reference data is then selected through the confidence information of land cover estimates and the status of use of high spatial resolution imagery provided by the metadata. Similarly to the procedural description used for fractional vegetation-soil compared to GLCVRD, we reclassify ten classes of this dataset into our four groups of endmembers, including (1) tree cover, shrub cover, herbaceous vegetation/grassland, cultivated and managed, and mosaic of cultivated and managed/natural vegetation to PV and NPV; (2) flooded/wetland and open water to DA; (3) urban and barren to BS; (4) snow and ice to IS. This involve comparing the measured percent of land cover with the mean of endmember fractions within the corresponding 1×1km pixels."

*5. The current analysis in the paper is predominantly focused on the Chinese region and its surroundings. Moreover, much of the work is centered around validating existing conclusions. Given that the authors have generated long-time-series global products, it is suggested to incorporate more globally relevant and newly discovered findings.*

Thank you for your insightful suggestions. Expanding the scope beyond the Chinese region and its surroundings to encompass more globally relevant insights aligns with the long-time-series products. We have therefore strengthened our analysis of the remaining regions in the Results and Discussion section.

Incorporating newly discovered findings on a global scale would enrich the depth and breadth of our analysis, and provide a more comprehensive understanding of Earth's dynamics. On one hand, we demonstrate the reliability of our data in surface process analysis by leveraging existing discoveries. On the other hand, our exploration of surface processes highlights the strengths of our data, wherein the interaction between these end-members enables a more accurate analysis of surface evolution. For instance, the simultaneous increase of photosynthetic and non-photosynthetic vegetation signifies the greening of the Earth driven by afforestation. These new findings are discussed in Section 4.2 Implications of global and regional shifts from pairs of two endmembers. Moreover, we'll diligently consider this recommendation to enhance the global relevance of our work and include other novel discoveries in our future researches (page 26, line 512-517).

"This dataset can serve as a baseline for enhancing our comprehension of heterogeneous surface dynamics and modeling Earth's biophysical processes through a multi-endmember coupling perspective, may significantly advance future research by serving as a foundational reference for delving deeper into complex land systems. Anticipating its potential applications across diverse domains such as ecology, climate studies, and urban planning, this dataset emerges as a pivotal resource. Its multifaceted utility is expected to play a pivotal role in informing environmental management decisions, advancing studies on ecological shifts, predicting climate

trends, and facilitating strategic landscape planning."

*6. The organizational structure of the paper needs revision. Currently, the Method, Results, and Discussion sections are intermingled with contents from other sections. For example, Lines 150-153, should be moved to the Introduction; Lines 312-313, Line 318-320 explain the further reasons, moving them to the Discussion should be better; Section 4.1 just describes the intercomparison of different products or methods, instead of discuss the potential reasons, I think it should be described in the Results section.*

Thank you for your detailed feedback regarding the organizational structure. We acknowledge the need for refinement and reorganization within the paper.

Wo have moved content describing the advantage of spectral mixture analysis model to the Introduction for better contextualization (page 4, line 94-96), and shifted the description of the comparison of different products or methods from Section 4.1 to the Section 3.2 to provide more clarity aligns with ensuring a logical flow within the paper.

We appreciate your input and will promptly address these structural adjustments to enhance the coherence and readability of our manuscript.

*Some technical problems:*

Thank you for your detailed feedback regarding the technical problems, we are sorry for these mistakes. we have corrected these errors and have also carefully examined and revised the manuscript to ensure the accuracy of the presentation.

*Lines 44-45: The full name of PV and NPV should be presented for the first time.*

■ Full name of PV and NPV has been presented (page 2, line 38-39).

*Line 72-73: indexes-->indices; leaf area index (LAI) should be a structural variable for the vegetation instead of vegetation index; The abbreviation for Normalized Difference Vegetation Index (NDVI) should be mentioned here.*

■ Continuous vegetation indexes (e.g., normalized difference vegetation index (NDVI), enhanced vegetation index (EVI)) provide limited information on surface composition, which hinders our ability of understanding ecosystem's structurally and functionally multifaceted shifts (page 3, line 72-74).

*Line 80: human -->human activity?*

■ under the influence of a changing environment and human activity (page 4, line 93)

*Lines 138-140: the introduction of the spectral mixture analysis model should be moved to the Introduction section.*

■ The introduction of the spectral mixture analysis model was blended into Introduction Section (page 4, line 94-96):

Recent studies have proven that spectral mixture analysis model has the advantage of providing more accurate and physically based representation of fraction vegetation-soil continues field in the subpixel level without training samples (Daldegan et al. 2019; Smith et al. 2007).

*Lines 153-155: You only mentioned the applicability of this classification system to tropical rainforests and drylands. Its suitability for global application is unclear.*

■ this framework was widely adopted for presentation surface structure worldwide, particularly in tropical rainforest and dryland ecosystems (page 6, line 167-168)

*Figure 2: The wavelength of B1-B7 should be specified.*

■ B1-B7 represent MODIS spectral bands, including 459-479nm, 545-565nm, 620-670nm, 841-876nm, 1230-1250nm, 1628-1652nm, and 2105-2155nm (page 10, line 221-222).

*Line 204: MESMA has already been declared in the Introduction, so it is not necessary to use its full name here.*

■ The MESMA has been used to estimate fractional vegetation-soil nexuses based on selected endmember spectra (page 10, line 224).

*Line 137 and Line 231: RMSE is defined twice. Different subscripts should be used for distinction.*

■ We have defined root-mean-square-error of MESMA fitting as $RMSE_{sma}$ (page 6, line 152-153).

*Lines 236-245: The introduction of the Mann-Kendall test should not be presented here, because the methods section should emphasize your own work. It is suggested to move it to the supporting information.*

■ We refined the descriptions of seasonal Mann-Kendall test (page 13-14, line 302-317).

*Line 262: "reasonably satisfactory" is not an object expression. Only presenting the numbers is enough here.*

■ Specifically, the performance of PV+NPV, BS, and IS endmember estimates have MAE less than 0.118, RMSE less than 0.149, $R^2$ greater than 0.592. (page 14, line 325-326).

*Lines 264-266 should be moved to the Discussion.*

■ We have move that to the Discussion.

*Line 274-277: Where is Figure 3 i-k? The full name of LAI should be mentioned here.*

■ We are sorry for this error; we have deleted that.

*Line 294: ×106 km2 -->×10⁶ km²*

■ $\times 10^6$ km² (page 20, line 393)

*Line 312: result from -->results from*

■ This outcome results from the forest loss induced soil exposure in Brazilian Amazon and Southeast Asia (page 20, line 411)

*Line 378: RESE -->RMSE*

■ It can be found that 90% of the $RMSE_{sma}$'s differences are concentrated within 1%. (page 18, line 371)

---

## Author Comment (AC3)

**Responses and Revisions to Comments**

**Dear reviewer**

Thanks for your reviewing and valuable comments of our manuscript entitled "***A global estimate of monthly vegetation and soil fractions from spatio-temporally adaptive spectral mixture analysis during 2001–2022***". We also appreciate you for providing insightful feedback and comments to strengthen our manuscript.

We have revised our manuscript with considering each detailed suggestion that you and reviewers have graciously provided. These major revisions include:

1 Added validation and comparison of the data.

2 Clearly defined the methodological workflow involving endmember selection, hierarchical clustering, validation, etc.

3 Analyzed and discussed the necessity and advantages of the data.

Besides these revisions, all authors checked the manuscript carefully and several minor revisions have been done to finalize the manuscript.

The following is a point-by-point response to the questions and comments delivered in your letter. For your convenience, revisions made by the authors have been highlighted in red color in the both response and revised manuscript, which could be easily checked. We hope that our revisions and responses can satisfactorily address all the issues and concerns.

*Sun et al. generated global monthly vegetation and soil fraction maps during 2001-2022 using a spectral mixture analysis method. Application of this method in global scale is interesting and deserves a scientific publication. However, the proposed product at current stage is not enough to publish on ESSD and the accuracy of this product is not very well demonstrated, from the perspective of a dataset and users. There are some main concerns:*

Thank you for your valuable feedback on our manuscript. We acknowledge your concerns regarding the reliability of the vegetation and soil fractions products described. We understand these concerns may stem from insufficiently validation in our manuscript. We have consequently enhanced the description of the validation process and added more validation/comparisons data.

On the one hand, we further authenticate our product through incorporating comprehensive global land cover and land use reference data, which were obtained from the Geo-Wiki crowdsourcing platform across four campaigns (page 12, line 265-276).

"Besides, we also authenticate our product through incorporating comprehensive global land cover and land use reference data (Fritz et al. 2017), which were obtained from the Geo-Wiki crowdsourcing platform across four campaigns: Human impact, wilderness, reference and disagreement. Over 150000 samples of land cover and land use were acquired in this reference data. To effectively validate our product, we need to filter the reference data, considering aspects such as data acquisition time, measurement methods, and credibility. We select first three campaigns, which have a good match with MODIS pixels (size 1×1km) and were observed during 2001 to 2022. High feasibility reference data is then selected through the confidence information of land cover estimates and the status of use of high spatial resolution imagery provided by the metadata. Similarly to the procedural description used for fractional vegetation-soil compared to GLCVRD, we reclassify ten classes of this dataset into our four groups of endmembers, including (1) tree cover, shrub cover, herbaceous vegetation/grassland, cultivated and managed, and mosaic of cultivated and managed/natural vegetation to PV and NPV; (2) flooded/wetland and open water to DA; (3) urban and barren to BS; (4) snow and ice to IS. This involve comparing the measured percent of land cover with the mean of endmember fractions within the corresponding 1×1km pixels."

Moreover, we strengthened the comparisons between our generated data and four existing datasets: NDVI, MOD44B Vegetation Continuous Fields product, GLASS FVC dataset, and GEOV Fcover dataset (page 12-13, line 277-301).

**"2.4 Comparisons and limitations analysis**

To verify the consistency and merits of our dataset against existing ones, we conducted comparisons with four distinct pre-existing datasets: NDVI, MOD44B Vegetation Continuous Fields product, GLASS fractional vegetation cover dataset, and GEOV Fcover dataset. NDVI is derived from monthly synthesized MCD43A4 images. Both mean values of NDVI and our estimated fractional PV across all years and months are considered for comparison. The MOD44B Vegetation Continuous Fields product provides annual information about the percent tree cover, percent non-tree cover, and percent non-vegetated within each 250-meter pixel globally (DiMiceli et al., 2015). Consequently, we compare vegetation cover proportions—sum of percent tree cover and percent non-tree cover—to the sum of fractional PV and NPV. To align spatial and temporal resolutions, we aggregated the sum of percent tree cover and percent non-tree cover to a 500-meter scale. Simultaneously, we computed monthly Fractional PV and NPV as annual averages. The GLASS fractional vegetation cover dataset, offering an 8-day temporal frequency and dual spatial resolutions of 0.05° and 500 meters, was generated using a machine learning approach correlating MODIS reflectance with fractional vegetation cover (Jia et al., 2015). In our study, the 500-meter GLASS data was utilized to validate our estimated fractions. We computed annual averages from all the CLASS fractional vegetation cover data within a year and compared it with the annual averages of Fractional PV and NPV. GEOV FCover is a 10-day product estimated through the neural network using visible, near-infrared and shortwave infrared at 1km resolution (Baret et al. 2013). We aggregate our product to a 1km spatial

resolution, and compare their annual averages with the annual averages of GEOV FCover.

Moreover, we also carry out a comparison with traditional linear spectral mixture analysis to demonstrate the advantages of our spatio-temporally adaptive spectral mixture analysis. Such comparison is performed using average of monthly $RMSE_{sma}$ of fully-constrained framework based on two fixed endmember spectral curves: (1) average of all spectral spectra for each endmember and (2) existing spectral spectra from Small and Sousa (2019).

Furthermore, to validate the uncertainties of the hierarchical clustering, we select a spectral spectrum from selected endmember spectra that exhibit the largest mean squared error from the mean of cluster for each cluster. These selected spectral spectra were then used to reconstruct an extreme library of endmember spectra and used to estimate fractional vegetation and soil using MESMA."

*1 The selection of endmember spectra is critical in this method. The authors establish a library of endmember spectral using a nested framework, combined with MODIS derived endmember spectra used in previous studies. However, if the proposed method is valid enough, why is it necessary to use those from previous studies? What are the differences between spectra decided in this study compared to existing ones? How about the proportion of endmember from your method and previous studies in the final library? Furthermore, the authors used a hierarchical clustering method to generate sub-groups of endmembers. This clustering method is quite dependent on the input parameters. Among a lot of clustering algorithm, why did you select this one? How about the input parameters, as well as the performance of its accuracy?*

First, we selected endmember spectra in ten globally representative regions using nested framework, and utilizing these previous spectra aimed to complement and enrich the diversity of the spectral library. In fact, we gather 7 PV, 5 NPV, 5 BS, and 1 DA endmember spectra through such literature search method, accounting for less than 15% of all endmember spectral curves.

We have revised this unclear description (page 8, line 205-207).

"Besides, we collect MODIS derived endmember spectra used in previous study to complement and enrich the diversity of the spectral library. (Okin et al. 2013; Daldegan et al. 2019; Meyer and Okin 2015; Sousa and Small 2019). We gather 7 PV, 5 NPV, 5 BS, and 1 DA endmember spectra through such literature search method."

Second, hierarchical clustering boasts strong interpretability and adaptability for clustering at diverse scales within data analysis[1]. Our use of hierarchical clustering aims to streamline the number of endmembers, thereby optimizing the efficiency of MESMA. This is crucial as MESMA requires intricate per-pixel adjustments, a challenge with
* * *
[1] Everitt, B. S., Landau, S., Leese, M., & Stahl, D. (2011). *Cluster analysis*. John Wiley & Sons.

extensive models due to current computational constraints.

In our approach, input parameters are all spectral curves per endmember to group similar curves to compute their mean—a representative typical spectral curve for each cluster. We analyze the uncertainties of such clustering in estimates of global vegetation and soil fractions, the results indicate the relative stability of the unmixed results as well as the effectiveness of the clustering.

We thus improved the descriptions of hierarchical clustering (page 8, line 210-214).

"To ensure feasibility of pixel-by-pixel operations in GEE, we also consider the similarity between the spectral curves, the hierarchical clustering method is selected to aggregate these spectra of each endmember as sub-groups, we input all spectral curves per endmember, grouping similar curves to compute their mean—a representative typical spectral curve for each cluster. Such hierarchical clustering boasts strong interpretability and adaptability for clustering at diverse scales within data analysis."

*2 Only five endmembers are select to represent the surface. More evidences are necessary to demonstrate the selection of five endmembers, e.g., group of current land classifications.*

Thank you for highlighting the importance of providing further evidence regarding the selection of five endmembers to represent the surface. Indeed, additional evidence, such as a comparison with current land classifications or other relevant groups, would bolster the rationale behind selecting these specific five endmembers.

(1) In fact, current research has demonstrated that the global land surface can essentially be characterized by four components—substrate, vegetation, dark and ice/snow— across diverse climate patterns and land cover types[2]. Moreover, to enrich the diversity of surface elements on a global scale, we also expanded the variety of endmember types based on endmember types commonly used in drylands [3]. These elements can characterize the fundamental composition of the Earth surface. Thus, we embody five endmembers to represent surface units.

We have improved the descriptions of five endmembers (page 6, line 163-171).

"(1) Recent studies have proposed various compositional endmember frameworks in different application contexts. For example, a framework including substrate, vegetation, dark and ice/snow was proposed and verified globally for both Landsat and MODIS to allow estimated fractions, this framework ensures consistent comparison of

[2] Sousa, D., Small, C.: Globally standardized MODIS spectral mixture models. Remote Sens Lett, 10, 1018-1027, https://doi.org/10.1080/2150704X.2019.1634299, 2019

[3] Guerschman, J. P., Scarth, P. F., McVicar, T. R., Renzullo, L. J., Malthus, T. J., Stewart, J. B., Trevithick, R.: Assessing the effects of site heterogeneity and soil properties when unmixing photosynthetic vegetation, non-photosynthetic vegetation and bare soil fractions from Landsat and MODIS data. Remote Sens. Environ., 161, 12-26, https://doi.org/10.1016/j.rse.2015.01.021, 2015

estimated fractions across diverse climate patterns and land cover types (Small and Milesi 2013; Sousa and Small 2019). Another framework includes photosynthetic vegetation, non-photosynthetic vegetation, soil, and shade (Roberts et al. 1993), this framework was widely adopted for presentation surface structure worldwide, particularly in tropical rainforest and dryland ecosystems (Guerschman et al., 2015). These elements can characterize the fundamental composition of the Earth surface. Thus, we embody five endmembers to represent surface units, these five endmembers include photosynthetic vegetation (PV), non-photosynthetic vegetation (NPV), bare soil (BS), dark (DA), ice/snow (IS)"

(2) Besides, Considering the scalability of hierarchical clustering and MESMA, we can analyze land cover types within the 15 subclasses under the five endmembers. We discover that these 15 subclasses encompassed tropical rainforest, temperate forest, cropland, grasslands, temperate deciduous forest in winter, crop residues, shrubs in dryland, moving sands, sand dunes, bare ground, moving sands. waters, bare rock, polar glaciers, and alpine glaciers. They cover major land cover types globally, indicating the representativeness of our selected endmembers (Fig. S2).

[Figure]

**Figure S2: Typical images representing selected pure pixel of each endmember**. a-o are tropical rainforest, temperate forest, cropland, grasslands, temperate deciduous forest in winter, crop residues, shrubs in dryland, moving sands, sand dunes, bare ground, moving sands. waters, bare rock, polar glaciers, and alpine glaciers.

(3) Furthermore, given sufficient computational resources, we can conduct more in-depth analyses and utilize MESMA for these 15 subclasses (Discussions, page 25, line 461-469).

"The spatio-temporally adaptive framework employed helps to increase the representativeness of endmember selection, and MESMA also considers the suitability of each combination of these endmembers within each pixel. However, considering the limitations of computational resources, our solution on hierarchical clusters of the endmember spectra can improve considerably cost-effective unmixing of long time-series satellite records over globe under the neglect of certain accuracy requirements (Fig. 3). With the assumption of increased computational power in the future, we believe that utilization of combination models from selected endmember spectra (35 GV spectra, 40 BS spectra, 25 NPV spectra, 16 DA spectra, and 15 IS spectra) or expanded endmember spectra may further improve the accuracy and stability of estimates of gradations of five surface vegetation and soil components at global scale."

*3 The validation step is very important to convince the users to use your datasets, instead of other existing products. However, the validation is not enough. For example, the authors used GLCVRD reference dataset which was produced in 2010 to validate the new products during the period during 2001-2022. How did you consider the inconsistency between time period and land changes during this long period? I would suggest the authors to improve the validation section, either by cross-validation with other existing products or by independent datasets.*

Due to challenges in conducting fraction estimation validation through field surveys, we employed validation methods based on high-resolution remote sensing imagery. The CLCVRD dataset is an example of such high-resolution reference data, hence we utilized it for corresponding validation. This dataset has demonstrated effectiveness in global-scale land cover verification. It's important to note that this dataset spans 2003-2012 (mainly 2010), and I apologize for any misleading information in previous descriptions. Thus, this data can effectively serve for fraction validation across different years and months.

(1) We have improved our description on CLCVRD and corresponding processes (page 11, line 236-251).

"Moreover, due to challenges in conducting fraction estimation validation through field surveys, we employ reference data obtained from high spatial resolution images as validation set. We thus select for two sets of reference data that their land cover classification systems are closely related to our five endmembers. Global Land Cover Validation Reference Dataset (GLCVRD) is provided with a 2m reference dataset from

very high resolution commercial remote sensing data within 5 × 5 km blocks from 2003 to 2012 (Olofsson et al. 2012; Pengra et al. 2015; Stehman et al. 2012). These datasets support global estimates of classification accuracy for four major land cover classes: tree, water, barren, other vegetation, cloud, shadow, ice & snow. Various recent studies have selected this dataset to evaluate the continuous fields of land cover types (Baumann et al. 2018; Qin et al. 2019; Song et al. 2018). We use all GLCVRD reference dataset (Fig. 3a) to assess the accuracy of globally fractional vegetation and soil estimates from MESMA. Firstly, we filter the estimated fractions based on the corresponding year and month obtained from the reference data. Simultaneously, aligning the interpretations of land cover types with our endmembers, we pair them accordingly, that is, tree and other vegetation represent PV and NPV, barren stands for BS, water and shadow correspond to DA, and ice & snow denote IS. Subsequently, we reclassify these paired land cover types and calculated their percentage within 5×5 km blocks, in which we exclude cloud coverage (named no data). Additionally, utilizing these cloud-free pixels in each block, we compute the mean of fractional values for each endmember, and then compare these estimated fractions with the measured percentage of paired the reclassified land cover types to validate the reliability of our product (Fig. S4)."

[Figure]

Figure S4: Procedural description for estimating fractional vegetation-soil compared to GLCVRD.

(2) Additionally, we conducted in-depth validation using new datasets[4] (page 12, line 265-276).

"Besides, we also authenticate our product through incorporating comprehensive global land cover and land use reference data (Fritz et al. 2017), which were obtained from the Geo-Wiki crowdsourcing platform across four campaigns: Human impact, wilderness, reference and disagreement. Over 150000 samples of land cover and land use were acquired in this reference data. To effectively validate our product, we need to filter the reference data, considering aspects such as data acquisition time, measurement methods, and credibility. We select first three campaigns, which have a good match with MODIS pixels (size 1×1km) and were observed during 2001 to 2022. High feasibility reference data is then selected through the confidence information of land cover estimates and the status of use of high spatial resolution imagery provided by the metadata. Similarly to the procedural description used for fractional vegetation-soil compared to GLCVRD, we reclassify ten classes of this dataset into our four groups of endmembers, including (1) tree cover, shrub cover, herbaceous vegetation/grassland, cultivated and managed, and mosaic of cultivated and managed/natural vegetation to PV and NPV; (2) flooded/wetland and open water to DA; (3) urban and barren to BS; (4) snow and ice to IS. This involve comparing the measured percent of land cover with the mean of endmember fractions within the corresponding 1×1km pixels."
* * *
[4] Fritz, S., See, L., Perger, C., McCallum, I., Schill, C., Schepaschenko, D., Duerauer, M., Karner, M., Dresel, C.,r Laso-Bayas, J. C., Lesiv, M., Moorthy, I., Salk, C. F., Danylo, O., Sturn, T., Albrecht, F., You, L., Kraxner F., Obersteiner, M.: A global dataset of crowdsourced land cover and land use reference data. Scientific data, 4, 1-8, https://doi.org/10.1038/sdata.2017.75, 2017.

[Figure]

**Fig. S5 Evaluation of global fractional endmember estimates based on land cover reference data.** a, the location of high-feasibility land cover reference data. b-d, Scatter plots of PV+NPV, BS, DA, IS fractions against land cover reference data

(3) Moreover, we strengthened the comparisons between our generated data and four existing datasets: NDVI, MOD44B Vegetation Continuous Fields product, GLASS FVC dataset, and GEOV Fcover dataset (page 12, line 277-301).

**"2.4 Comparisons and limitations analysis**

To verify the consistency and merits of our dataset against existing ones, we conducted comparisons with four distinct pre-existing datasets: NDVI, MOD44B Vegetation Continuous Fields product, GLASS fractional vegetation cover dataset, and GEOV Fcover dataset. NDVI is derived from monthly synthesized MCD43A4 images. Both mean values of NDVI and our estimated fractional PV across all years and months are

considered for comparison. The MOD44B Vegetation Continuous Fields product provides annual information about the percent tree cover, percent non-tree cover, and percent non-vegetated within each 250-meter pixel globally (DiMiceli et al., 2015). Consequently, we compare vegetation cover proportions—sum of percent tree cover and percent non-tree cover—to the sum of fractional PV and NPV. To align spatial and temporal resolutions, we aggregated the sum of percent tree cover and percent non-tree cover to a 500-meter scale. Simultaneously, we computed monthly Fractional PV and NPV as annual averages. The GLASS fractional vegetation cover dataset, offering an 8-day temporal frequency and dual spatial resolutions of 0.05° and 500 meters, was generated using a machine learning approach correlating MODIS reflectance with fractional vegetation cover (Jia et al., 2015). In our study, the 500-meter GLASS data was utilized to validate our estimated fractions. We computed annual averages from all the CLASS fractional vegetation cover data within a year and compared it with the annual averages of Fractional PV and NPV. GEOV FCover is a 10-day product estimated through the neural network using visible, near-infrared and shortwave infrared at 1km resolution (Baret et al. 2013). We aggregate our product to a 1km spatial resolution, and compare their annual averages with the annual averages of GEOV FCover.

Moreover, we also carry out a comparison with traditional linear spectral mixture analysis to demonstrate the advantages of our spatio-temporally adaptive spectral mixture analysis. Such comparison is performed using average of monthly $RMSE_{sma}$ of fully-constrained framework based on two fixed endmember spectral curves: (1) average of all spectral spectra for each endmember and (2) existing spectral spectra from Small and Sousa (2019).

Furthermore, to validate the uncertainties of the hierarchical clustering, we select a spectral spectrum from selected endmember spectra that exhibit the largest mean squared error from the mean of cluster for each cluster. These selected spectral spectra were then used to reconstruct an extreme library of endmember spectra and used to estimate fractional vegetation and soil using MESMA."

[Figure]

**Figure 4: Comparisons with other datasets and traditional spectral mixture analysis models. a, b, c, d** the bi-dimensional histogram of fractional endmembers and other dataset with bin size of 2%, including fractional PV against NDVI (a), fractional PV and NPV against fractional tree and non-tree vegetation of MOD44B vegetation continuous fields product (b), fractional PV and NPV against GLASS fractional vegetation cover product (c), fractional PV and NPV against fractional vegetation cover of GEOV Fcover product; **e, f**, the boxplot and violin plot for average of monthly $RMSE_{sma}$ for two fixed endmember spectral curves using fully constrained linear spectral mixture models, including (e) average of all spectral spectra for each endmember and (f) existing spectral spectra from Small and Sousa (2019).

[Figure]

**Fig. S6 The detailed graphs for comparing different datasets**. a, b, and c represent comparisons of vegetation abundance products in different scenarios, specifically, regions with low vegetation cover in arid areas, high vegetation cover in tropical rainforests, and transitional zones from low to high values. The compared products include our produced PV and NPV, MOD44B, GLASS, and GEOV.

*4 The structure of this manuscript should be re-organized, especially the results and discussion. Put all figures in results section. In discussion section, I expect more deep analysis and discussions on the intercomparison with existing products, the advantages and disadvantages of the new products, and uncertainties.*

Thank you for your valuable feedback on the manuscript's structure, particularly

emphasizing the need for reorganization, especially in the Results and Discussion sections. We will consolidate some figures within the Results section for improved coherence. In the Discussion section, we aim to delve deeper into comprehensive analyses and discussions. We'll focus on intricate intercomparisons with existing products, elucidating the advantages, disadvantages, and uncertainties inherent in the new products. This restructured discussion will offer a more comprehensive and insightful exploration of our findings, ensuring a clearer and more informative narrative.

We have moved "Compared with other datasets and traditional SMA model" and "Uncertainties of estimates of global vegetation and soil fractions" to Section 3.2 and Section 3.3. And we have improved the discussions of Advances and limitations of estimates of global vegetation and soil fractions in Section 4.1.

*5 Some minor comments:*

Thank you for your detailed feedback regarding the technical problems, we have corrected these errors and have also carefully examined and revised the manuscript to ensure the accuracy of the presentation.

*Line 85-86: this is not the key scientific question of this study. More emphasis should be started from issues of existing global vegetation/soil fraction products, instead of application of spectral mixture analysis method in global scale.*

■ We have improved descriptions of existing global vegetation/soil fraction products (page 3, line 75-87).

"In recent years, there have been significant advancements in fractional vegetation cover within the fields of remote sensing and environmental science. This progress has led to the development of various products at multiple resolutions, such as long-term global land surface satellite (GLASS), GEOV Fcover, multi-source data synergized quantitative remote sensing production system (MuSyQ) fractional vegetation cover (Baret et al. 2013; Jia et al., 2015; Mu et al., 2017; Zhao et al., 2023). These products primarily integrate and utilize data from different spectral bands and sensors, employing methods including machine learning and radiative transfer model. However, these data primarily focus on green vegetation, posing significant limitations in capturing information regarding non-photosynthetic vegetation and bare soil. This constraint also restricts the applicability of this data in arid regions. Although some initiatives and products focused on multi-element fractions, such as MOD44B and the Global Vegetation Fractional Cover Product (DiMiceli et al., 2015; Guerschman et al., 2015). For instance, the Global Vegetation Fractional Cover Product primarily targets arid regions, particularly Australia, focusing on photosynthetic vegetation, non-photosynthetic vegetation, and bare soil. Meanwhile, MOD44B achieves global-scale acquisition of trees, non-trees, and non-vegetative cover. There is a lack of unified classification systems among these products across global scale"

*Line 135: Spectral mixture analysis, 's' should be capitalized.*

■ We have improved as "Spectral mixture analysis"

*Line 166: what is the resolution of this land cover product? There are multiple land cover products, why do you select this one? How did you overlap the climate classification zones and the land cover products when the resolution is quite different?*

■ MCD12Q1 is a 500m land use cover data from Terra and Aqua Version 6 product. We selected the MCD12Q1 land cover product due to its widely acknowledged accuracy, high spatial resolution, and global coverage. This product is generated by a robust algorithm using multiple remote sensing datasets, providing comprehensive and detailed land cover information across various land cover classes. Its consistency, reliability, and compatibility with our study's objectives make it an ideal choice for comparison and validation against our newly developed products.

Moreover, we do not overlap the climate classification zones and the land cover products, we used MODIS sinusoidal grid ($10° \times 10°$ intervals) to count the land cover diversity (D) and the full coverage capacity of climate types.

"Finally, we selected the top 10 grids (i.e., h08v05, h12v12, h13v09, h16v01, h21v03, h22v02, h22v08, h24v06, h26v05, h27v06, h29v12) in terms of Simpson's Diversity Index (D) among all MODIS grids (Fig. S1a, b), and containing all Köppen-Geiger climate types" (page 7, line 182-185)

*Line 180-181: While \*\*\*, especially in vegetation growing seasons. This sentence seems not complete.*

■ We have revised that as "PC eigenvector with relatively high contrast between the near-infrared band and other bands primarily captures information related to photosynthetic vegetation (PV), particularly during vegetation growing seasons. (page 7, line 193-195)"

*Line 192: 35 GV spectra, should be PV spectra?*

■ We establish a library of endmember spectra considering spatio-temporal variability, this library includes 35 PV spectra, 40 BS spectra, 25 NPV spectra, 16 DA spectra, and 15 IS spectra (page 8, line 208).

*Lien 212-213: how long it took to generate one global map on GEE?*

■ Exporting the data will take approximately 30-40 minutes per month per our generated grid, according to the grid of longitude 60° and Latitude 50°.

*Line 260: It is strange that Sahara Desert and polar regions had higher RMSE. These regions were well-known for unique land cover type.*

- Even though the two regions have a single land cover type of desert and snow/ice, the internal spectral profile is more complex and extreme, for example, deserts have both high and low reflectance sands. However, to realize the global-scale dominant element information extraction, we input all spectral curves per endmember, grouping similar curves to compute their mean, resulting in larger $RMSE_{sma}$ for these extreme regions, but basically, they are also in line with the requirements of model fitting accuracy.

  Such inaccurately pixels have been discussed in Section 4.1 Advances and limitations of estimates of global vegetation and soil fractions (page 25, line 481-485).

*Figure 3.e-h are not enough to support your conclusion.*

- We conducted in-depth validation using new datasets and discussed the limitations of validation using current land cover reference dataset (see details in response of comment 3).

*Section 3.1: add more analysis for pixels that were estimated inaccurately.*

- We have added more discissions for pixels that were estimated inaccurately (page 25, line 481-485).

  "We observed higher $RMSE_{sma}$ values in seemingly homogeneous areas like the Sahara Desert and Arctic regions. However, within these regions, there often exist extremely diverse land cover types, such as high and low reflectance sands and ice. When selecting endmembers and hierarchical clustering models, we might not have adequately considered these extreme spectral curves. As a result, these extreme areas exhibit a higher uncertainty."

*Lie 358: indicate the resolution of MODIS pixel.*

- each MODIS pixel (500 m) (page 24, line 434)

---

## Author Comment (AC4)

**Revisions and Responses to Comments**

**Dear reviewer**

Thanks for your reviewing and valuable comments of our manuscript entitled "***A global estimate of monthly vegetation and soil fractions from spatio-temporally adaptive spectral mixture analysis during 2001–2022***". We also appreciate you for providing insightful feedback and comments to strengthen our manuscript.

We have revised our manuscript with considering each detailed suggestion that you and reviewers have graciously provided. These major revisions include:

1 added additional validation and comparison data

2 provided a more accurate description to enhance the clarity and coherence of the presentation.

Besides these revisions, all authors checked the manuscript carefully and several minor revisions have been done to finalize the manuscript.

The following is a point-by-point response to the questions and comments delivered in your letter. For your convenience, revisions made by the authors have been highlighted in red color in the both response and revised manuscript, which could be easily checked. We hope that our revisions and responses can satisfactorily address all the issues and concerns.

*This paper aims to provide a globally comprehensive record of monthly vegetation and soil fractions during the period 2001–2022 using the spatiotemporally adaptive MESMA methods at the powerful Google Earth Engine (GEE) platform. However, in my view, some issues should be resolved. Please find my detailed comments below.*

Thanks for your reviewing and valuable comments of our manuscript, we have revised our manuscript with considering each detailed suggestion that you and reviewers have graciously provided.

*The overall structure of this article appears to be somewhat confusing. I recommend considering adjustments to enhance the clarity and coherence of the presentation.*

Thank you for your valuable feedback on the manuscript's structure, particularly emphasizing the need for reorganization. We have moved "Compared with other datasets and traditional spectral mixture analysis model" and "Uncertainties of estimates of global vegetation and soil fractions" to Section 3.2 and Section 3.3. In the Discussion section, we have improved the discussions of Advances and limitations of estimates of global vegetation and soil fractions in Section 4.1 to delve deeper into comprehensive analyses and discussions.

*In your paper, you have utilized Köppen-Geiger climate classification maps Version 1, while Version 2 is also available at https://www.gloh2o.org/koppen/. Could you please provide some insights into the rationale behind selecting Version 1 over Version 2? Additionally, considering the abundance of global land cover products, what motivated the choice of MCD12Q1?*

We opted for Köppen-Geiger climate classification maps Version 1 due to their widespread acceptance and usage within the scientific community at the period of our research. While Version 2 is available and might offer certain improvements or updates, our choice was based on the prevalence and familiarity of Version 1 within the research community during our study period. Moreover, we utilized this data solely to ensure that our selected representative MODIS grids encompass all climate zones. Therefore, the utilization of data from versions 1 and 2 had negligible influence.

Regarding the selection of MCD12Q1 among various global land cover products, we chose it due to its established accuracy, high spatial resolution, and comprehensive coverage. MCD12Q1 utilizes multiple datasets and robust algorithms to provide detailed and reliable land cover information, aligning well with the requirements of our study for endmember selection. Its proven consistency and compatibility with our research objectives influenced our decision in utilizing this specific product.

We have improved descriptions of Köppen-Geiger climate classification (page 5, line 124-126) and MCD12Q1 (page 5, line 132-135).

"The Köppen-Geiger climate classification is a reasonable approach to aggregate complex climate gradients into a simple but ecologically meaningful classification scheme (Beck et al. 2018). This dataset presents their widespread acceptance and usage within the scientific community."

"MCD12Q1 utilizes multiple datasets and robust algorithms, and provides detailed and reliable land cover information. It has been proven advantages in representing the global land cover structure, patterns, and dynamics, aligning well with the requirements of our study for endmember selection."

*In this paper, authors selected the typical sites employed for standardized endmembers selection based on 2020 MCD12Q1. Could you elaborate on the decision-making process behind employing this particular year for standardized endmembers selection?*

Thank you very much for your suggestion. We chose the MCD12Q1 data not directly for endmember selection but for identifying representative zones. Subsequently, we will carry out endmember selection in these representative regions. The criterion for selecting representative MODIS grids is a high diversity of land cover types. Therefore, we need to choose the recent available MCD12Q1 land cover dataset (in 2020) to calculate land cover diversity (page 7, line 179-180).

"Meanwhile, we also examine land cover diversity, characterized by Simpson's Diversity Index (D) of recent MCD12Q1 Version 6 product in 2020 in each MODIS grid"

**When comparing your estimated vegetation and soil fractions dataset, along with NDVI, fractional PV, and NPV, against the MOD44B vegetation continuous fields product, I noticed that you specifically chose to focus on the tree and non-tree vegetation components. Could you elaborate on the reasoning behind this selective comparison?**

We are aware that MOD44B covers only three components: percent tree cover, percent non-tree cover, and percent non-vegetated (bare), whereas our endmember types encompass five. We thus chose to combine tree and non-tree vegetation components as vegetated area, and then it was compared with our PV and NPV due to considerations of consistency across different classification systems.

We have refined the descriptions of comparisons (page 12-13, line 278-293).

"To verify the consistency and merits of our dataset against existing ones, we conducted comparisons with four distinct pre-existing datasets: NDVI, MOD44B Vegetation Continuous Fields product, GLASS fractional vegetation cover dataset, and GEOV Fcover dataset.   NDVI is derived from monthly synthesized MCD43A4 images. Both mean values of NDVI and our estimated fractional PV across all years and months are considered for comparison. The MOD44B Vegetation Continuous Fields product provides annual information about the percent tree cover, percent non-tree cover, and percent non-vegetated within each 250-meter pixel globally (DiMiceli et al., 2015). Consequently, we compare vegetation cover proportions—sum of percent tree cover and percent non-tree cover—to the sum of fractional PV and NPV. To align spatial and temporal resolutions, we aggregated the sum of percent tree cover and percent non-tree cover to a 500-meter scale. Simultaneously, we computed monthly Fractional PV and NPV as annual averages. The GLASS fractional vegetation cover dataset, offering an 8-day temporal frequency and dual spatial resolutions of 0.05° and 500 meters, was generated using a machine learning approach correlating MODIS reflectance with fractional vegetation cover (Jia et al., 2015). In our study, the 500-meter GLASS data was utilized to validate our estimated fractions. We computed annual averages from all the CLASS fractional vegetation cover data within a year and compared it with the annual averages of Fractional PV. GEOV FCover is a 10-day product estimated through the neural network using visible, near-infrared and shortwave infrared at 1km resolution (Baret et al. 2013). We aggregate our product to a 1km spatial resolution, and compare their annual averages with the annual averages of GEOV FCover."

*Could you please provide more detailed reasons or methods explaining why you decided to verify PV and NPV as a single category in your study? Additionally, I noticed that you exclusively chose GLCVRD reference data from 2010 for validation*

*purposes. Could you please elaborate on the decision-making process behind selecting this specific year for validation?*

As mentioned in the previous response, there isn't currently a suitable product available to validate photosynthetic and non-photosynthetic vegetation independently. Therefore, in our validation process, we conducted validation using vegetation as a combined category.

Moreover, the CLCVRD dataset has demonstrated effectiveness in global-scale land cover verification. It's important to note that this dataset spans 2003-2012 (mainly 2010), and I apologize for any misleading information in previous descriptions. Thus, this data can effectively serve for fraction validation across different years and months.

We thus improved the descriptions of validations (page 11 line 244-251) and discussed the limitations of validation datasets (page 25, line 470-481).

"Firstly, we filter the estimated fractions based on the corresponding year and month obtained from the reference data. Simultaneously, aligning the interpretations of land cover types with our endmembers, we pair them accordingly, that is, tree and other vegetation represent PV and NPV, barren stands for BS, water and shadow correspond to DA, and ice & snow denote IS. Subsequently, we reclassify these paired land cover types and calculated their percentage within 5×5 km blocks, in which we exclude cloud coverage (named no data). Additionally, utilizing these cloud-free pixels in each block, we compute the mean of fractional values for each endmember, and then compare these estimated fractions with the measured percentage of paired the reclassified land cover types to validate the reliability of our product (Fig. S4)."

[Figure]

**Figure S4: Procedural description for estimating fractional vegetation-soil compared to GLCVRD.**

"However, due to the absence of corresponding reference data for validation, we solely rely on two high-quality land cover reference datasets for validation. Unfortunately, these datasets do not intricately characterize small-scale shadows and bare soil within complex vegetation structures. Consequently, this leads to a misconception in the validation, where our DA and BS are overestimated in low-value areas and vegetation is underestimated in high-value areas (Fig. 3, Fig.S5). Therefore, in the future, there is a need to further develop high-quality relevant reference data. Considering that MOD44B vegetation continuous fields product provides a gradation of three surface cover components: percent tree cover, percent non-tree cover, and percent bare, the dark components (i.e., shadow of vegetation and mountain, water) are not quantified. Therefore, fractional PV and NPV is overall biased high, especially in areas with PV and NPV less than 0.50 (Fig. 4b; Fig. S6b). Besides, we also observed a certain degree of underestimation in these three datasets in regions with lower vegetation cover compared to our data. This is mainly because these datasets focus solely on green vegetation, especially GLASS and GEOV Fcover (Baret et al. 2013; Jia et al., 2015), and do not accurately estimate non-photosynthetic vegetation in arid regions. The above comparisons demonstrate our precision advantage in fine extraction of multiple endmembers."

*Given that your product has a spatial resolution of 500m and a monthly temporal resolution, while the MOD44B product has a spatial resolution of 250m and a yearly temporal resolution, could you please provide a detailed description of the comparison process and the methods employed between them?*

Certainly, the comparison process between our product with a 500m spatial resolution and monthly temporal resolution and the MOD44B product with a 250m spatial resolution and yearly temporal resolution involved several steps and considerations.

To conduct the comparison, we first aggregated the monthly data into a yearly format to align with the temporal resolution of MOD44B. Meanwhile, we aggregated 250m MOD44B product to our data to match the spatial resolutions of our products. This allowed us to bridge the temporal gaps and ensure a meaningful evaluation despite the differing temporal frequencies.

We have improved such descriptions of comparisons (page 12-13, line 278-286).

"To verify the consistency and merits of our dataset against existing ones, we conducted comparisons with four distinct pre-existing datasets: NDVI, MOD44B Vegetation Continuous Fields product, GLASS fractional vegetation cover dataset, and GEOV Fcover dataset.   NDVI is derived from monthly synthesized MCD43A4 images. Both mean values of NDVI and our estimated fractional PV across all years and months are considered for comparison. The MOD44B Vegetation Continuous Fields product

provides annual information about the percent tree cover, percent non-tree cover, and percent non-vegetated within each 250-meter pixel globally (DiMiceli et al., 2015). Consequently, we compare vegetation cover proportions—sum of percent tree cover and percent non-tree cover—to the sum of fractional PV and NPV. To align spatial and temporal resolutions, we aggregated the sum of percent tree cover and percent non-tree cover to a 500-meter scale. Simultaneously, we computed monthly fractional PV and NPV as annual averages."

*Please revise the abstract of this paper, some sentences are confusing. For example, line 32-34: "Sustainably managing terrestrial ecosystems requires an increased understanding of these structurally and functionally heterogeneous multi-component information and their changes, but we remain lack of such records of fractional vegetation and soil information at global scale.*

Certainly! Here's a revised version for the mentioned section (page 2 line 32-35):

"Sustainably managing terrestrial ecosystems necessitates a deeper comprehension of the diverse and dynamic nature of multi-component information within these environments. However, comprehensive records of global-scale fractional vegetation and soil information that encompass these structural and functional complexities remain limited."

*Abbreviations are preferably not used in the abstract of a paper, and if they are used in the abstract, the full name of each abbreviation should be presented the first time it appears. For example, Lines 44-45: "PV and NPV".*

Thank you very much for your suggestion. We have revised the abbreviations from the Abstract (page 2 line 38-39).

"five physically meaningful vegetation and soil endmembers, including photosynthetic vegetation (PV), non-photosynthetic vegetation (NPV), bare soil (BS), ice/snow (IS), and dark surface (DA), with high accuracy and low uncertainty"

*Line 124: Please correct "croplands and mosaics of croplands and natural vegetation;" to "Cropland/Natural Vegetation Mosaics".*

Thank you very much for your suggestion, we have revised "croplands and mosaics of croplands and natural vegetation" to "Cropland/Natural Vegetation Mosaics" (page 5 line 139)

*Line 137 and 231: Please add the serial number to the formula in this paper. For example, RMSE was defined twice.*

Thank you for your thorough review of our manuscript. We have redefined the Root Mean Square Error for the fitting of the MESMA model as $RMSE_{sma}$. Additionally, we have added numbering to the formulas (page 6 line 152-153).

*Line 171: Why did the authors choose the threshold 'D > 0.7' in this study, and are there any references supporting this decision?*

The threshold selection was based on ranking the Simpson's Diversity Index (D) of all MODIS grids. We chose the top ten grids, hence setting the threshold at 0.7. To avoid confusion, we rephrased it as (page 7 line 182-184),

"Finally, we selected the top 10 grids (i.e., h08v05, h12v12, h13v09, h16v01, h21v03, h22v02, h22v08, h24v06, h26v05, h27v06, h29v12) in terms of Simpson's Diversity Index (D) among all MODIS grids (Fig. S1a, b)"

*Line 268-277: Could you please provide the location or clarification for Figure 3 i-k? I would like to ensure that all relevant chart information is accurate and accessible.*

We sincerely apologize for the error in our expression. We have made the necessary corrections and also reviewed all the titles of the figures and tables

*Line 331 and 348: The subtitle '4.1 Compared with other datasets and traditional SMA model' seems inconsistent with 'Figure 7: Comparisons with other datasets and LSMA models.' Could you please clarify whether the LSMA model is considered one of the traditional SMA models, and ensure consistency in the expression used?*

Thank you for pointing this out. We have revised unclear expression to ensure consistency between the LSMA model and traditional SMA models in the context of the comparisons made in Figure 4. The traditional SMA models was defined as fully constrained linear spectral mixture models (page 17, line 363-369) .

"**Figure 4: Comparisons with other datasets and traditional spectral mixture analysis models. a, b**, **c**, **d** the bi-dimensional histogram of fractional endmembers and other dataset with bin size of 2%, including fractional PV against NDVI (a), fractional PV and NPV against fractional tree and non-tree vegetation of MOD44B vegetation continuous fields product (b), fractional PV and NPV against GLASS fractional vegetation cover product (c), fractional PV and NPV against fractional vegetation cover of GEOV Fcover product; **e**, **f**, the boxplot and violin plot for average of monthly $RMSE_{sma}$ for two fixed endmember spectral curves using fully constrained linear spectral mixture models, including (e) average of all spectral spectra for each endmember and (f) existing spectral spectra from Small and Sousa (2019)."

*Line 378: "RESE" --> "RMSE".*

We sincerely apologize for the error in our expression. We have made the corrections. All authors checked the manuscript carefully and several minor revisions have been

done to finalize the manuscript.

---

## Referee Report (RR1)

This is an extensive revision. It reads much better. Thanks for addressing my comments. The accuracy of this product remains a major concern, and I don't think that has been well demonstrated.

The advantage of this dataset is offering various surface fractions. I thus urge the authors to compare green vegetation fractional cover (or fractional NPV, not the sum of fractional PV and NPV in Figures 3, 4, and S5) of this dataset vs. that of present products. Especially in the evaluation of the accuracy of fractional NPV, because knowledge of NPV is essential for all terrestrial ecosystems, and its fraction estimates are a highlight of this work.

Lines 79-80: The dimidiate pixel model is also important for satellite-derived FVC products.

Lines 80-81: The meaning of NPV, which is considered a key quantifiable variable for terrestrial ecosystems, should be supplemented to emphasize the value of this work.

---

## Author Response (AR2)

**Responses and Revisions to Comments**

**Dear reviewer**

Thanks for your reviewing and valuable comments of our manuscript entitled "***A global estimate of monthly vegetation and soil fractions from spatio-temporally adaptive spectral mixture analysis during 2001–2022***". We also appreciate you for providing insightful feedback and comments to strengthen our manuscript.

We have revised our manuscript with considering each detailed suggestion that you have graciously provided. And we have enhanced the accuracy validation of the product.

The following is a point-by-point response to the questions and comments delivered in your letter. For your convenience, revisions made by the authors have been highlighted in red color in the both response and revised manuscript, which could be easily checked. We hope that our revisions and responses can satisfactorily address all the issues and concerns.

**Referee #1**

*This is an extensive revision. It reads much better. Thanks for addressing my comments. The accuracy of this product remains a major concern, and I don′t think that has been well demonstrated.*

*The advantage of this dataset is offering various surface fractions. I thus urge the authors to compare green vegetation fractional cover (or fractional NPV, not the sum of fractional PV and NPV in Figures 3, 4, and S5) of this dataset vs. that of present products. Especially in the evaluation of the accuracy of fractional NPV, because knowledge of NPV is essential for all terrestrial ecosystems, and its fraction estimates are a highlight of this work.*

Thank you very much for your valuable suggestions, and we agree that the estimation of NPV fractions is one of the key highlights of this paper. This product offering surface fractional PV and NPV is important for accurately characterizing vegetation structure. Therefore, we are in favor of your suggestion to validate and compare PV and NPV abundance values with that of present products, separately.

(1) We have already carried out a comparison of PV and NDVI in Figure 4a. The results indicate a strong positive relationship between PV fraction and NDVI ($p<0.01$). Yet, this correlation becomes less pronounced when PV exceeds 50%, suggesting an evident saturation effect within NDVI.

(2) We also included an experiment of comparing LAI, a parameter that characterizes the structure of green foliage, with our estimated fractions of PV (page 12, line 284-285; page 16, line 353-356). The comparison results further demonstrated that linear relationship also exists in the relationship between PV and LAI, but a nonlinear turning point occurs when PV exceeds 70% (Fig. 4c). Such relationship is generally consistent with the results of previous research on the relationship between LAI and FVC, which manifests as a broom shape with distinct confidence interval values distributed along the FVC gradients[1]

[Figure]

**Figure 4: Comparisons with other datasets and traditional spectral mixture analysis models. a, b, c, d, e,** the bi-dimensional histogram of fractional endmembers and other dataset with bin size of 2%, including fractional PV against NDVI (a), fractional PV and NPV against fractional tree and non-tree vegetation of MOD44B vegetation continuous fields product (b), fractional PV against LAI (c), fractional PV and NPV against GLASS fractional vegetation cover product (d), fractional PV and NPV against fractional vegetation cover of GEOV Fcover product (e); **f**, **g**, the boxplot and violin plot for average of monthly $RMSE_{sma}$ for two fixed endmember spectral curves using fully constrained linear spectral mixture models, including (e) average of all spectral spectra for each endmember and (f) existing spectral spectra from Small and Sousa (2019).

(3) Due to the lack of suitable existing datasets, it is difficult for us to compare NPV with existing data products. But we have already demonstrated the mapping validity from detailed spatial distribution of NPV in typical regions, when we compared that to Google Earth images (Fig. S6). The current lack of fractional NPV products highlights the need for ongoing development in our future work, and more detailed comparison and validation experiments of NPV are needed for improving the

[1] Fang, H., Li, S., Zhang, Y., Wei, S., & Wang, Y. (2021). New insights of global vegetation structural properties through an analysis of canopy clumping index, fractional vegetation cover, and leaf area index. Science of Remote Sensing, 4, 100027.

accuracy, feasibility and validity. This view has been highlighted in Discussion (page 24, line 490-494).

"Given the importance of NPV in ecological research, undertaking separate validation and comparisons between PV and NPV represents a critical foundational effort. While detailed maps of a representative region illustrate the reliability and advantages of our NPV estimation over other products (Fig. S6), the current lack of equivalent products highlights the need for ongoing development. Enhancing quantitative comparison efforts will be essential to bolster the feasibility, accuracy and validity of our NPV product in future studies."

*Lines 79-80: The dimidiate pixel model is also important for satellite-derived FVC products.*

Thank you very much for your suggestion. We have added information about the progress in dimidiate pixel model and relevant literature in the Introduction section (page 3, line 79-81).

"These products primarily integrate and utilize data from different spectral bands and sensors, employing methods including machine learning, radiative transfer model and dimidiate pixel model (Baret et al. 2013; Yan et al., 2021; Zhao et al., 2023)."

*Lines 80-81: The meaning of NPV, which is considered a key quantifiable variable for terrestrial ecosystems, should be supplemented to emphasize the value of this work.*

Thanks for your valuable, we agree that NPV (Non-Photosynthetic Vegetation) is a crucial highlight of this work. Therefore, we have provided a detailed description and introduction of NPV in the Introduction Section, outlining its significant role in ecological research and the value of our work(page 3, line 82-86).

"In ecological studies and remote sensing, non-photosynthetic vegetation including stems, branches, and other plant structures primarily serve ecosystem functions other than photosynthesis, such as support and storage. Therefore, understanding the distribution and characteristics of non-photosynthetic vegetation is important for a comprehensive analysis of ecosystems and land cover, especially in drylands (Guerschman et al., 2009)."

We also improved discussions of NPV in Discussion Section to emphasize the value of this work (page 21, line 458-461)

"The NPV component is a vital element in arid ecosystems and represents a crucial part of vegetation biomass. Our dataset, by finely characterizing NPV, not only aids in understanding the evolving features of vegetation structure under photosynthetic and non-photosynthetic interactions (Guerschman et al. 2015), but also contributes to a more accurate quantification of global biomass in arid land systems (Smith et al. 2019)."

**Referee #3**

*I appreciate the author's effort in incorporating my comments and suggestions. Overall, my concerns were well organized and addressed.*

Thank you for acknowledging the efforts to incorporate your comments and suggestions. I'm glad to hear that your concerns were well organized and addressed.

**Referee #4**

*Dear authors, After checking the response as well as the revised manuscript, I think this paper can be accepted now. Thanks!*

Thank you for reviewing the response and the revised manuscript. We appreciate your positive feedback and are pleased to hear that the paper can now be accepted. Your input has been valuable in improving the quality of the manuscript.